# DexMan: Learning Bimanual Dexterous Manipulation from Human and Generated Videos

## Abstract

We present DexMan, an automated framework that converts human visual demonstrations into bimanual dexterous manipulation skills for humanoid robots in simulation. Operating directly on third-person videos of humans manipulating rigid objects, DexMan eliminates the need for camera calibration, depth sensors, scanned 3D object assets, or ground-truth hand and object motion annotations. Unlike prior approaches that consider only simplified floating hands, it directly controls a humanoid robot and leverages novel contact-based rewards to improve policy learning from noisy hand–object poses estimated from in-the-wild videos. DexMan achieves state-of-the-art performance in object pose estimation on TACO, with absolute gains of 0.08 and 0.12 in ADD-S and VSD. Meanwhile, its RL policy surpasses previous methods by 19% success rate on OakInk-v2. Furthermore, DexMan can generate skills from both real and synthetic videos, without the need for manual data collection and costly motion capture, and enabling the creation of large-scale, diverse datasets for training generalist dexterous manipulation. Video results are available on our webpage: dexman2026.github.io

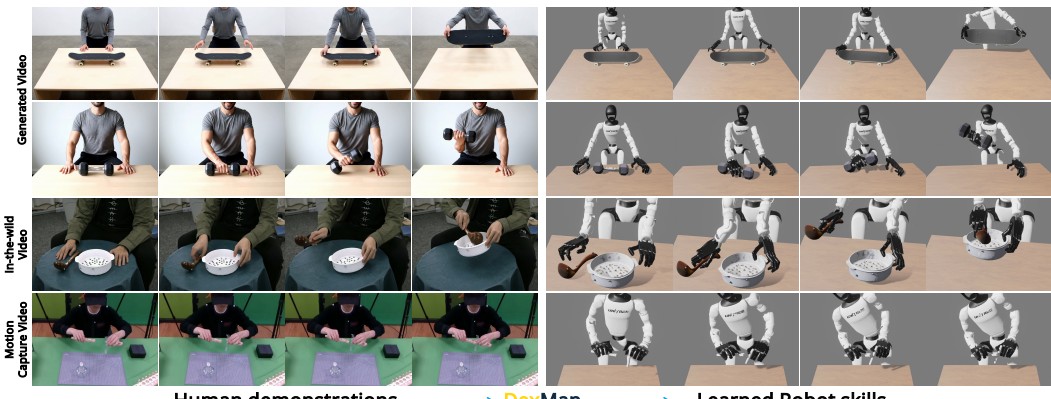

Figure 1: DexMan is an automated framework that transfers human visual demonstrations into bimanual dexterous manipulation skills for humanoid robots in simulation. Going beyond motion-capture data, DexMan can generate skills from either in-the-wild or synthetic videos, eliminating the need for manual data collection, and thereby enabling the curation of large-scale robotic datasets.

## 1 Introduction

To handle dynamic and contact-rich interactions involved in daily manipulation tasks, robots must coordinate multiple arms and dexterous fingers. However, existing learning frameworks, such as behavior cloning (BC) (Lin et al., 2024; Wang et al., 2024) and reinforcement learning (RL) (Chen et al., 2022a; Wan et al., 2023), struggle with the increased dimensionality of multi-arm, multi-fingered manipulators. This complexity raises the cost of collecting teleoperated demonstrations for BC (Lin et al., 2024; Wang et al., 2024) and amplifies inherent sample inefficiency of RL (Chen et al., 2022a; Wan et al., 2023), making it difficult to train effective policies for multi-limb robots.

The morphological similarity to humans motivates learning from human demonstrations (Pollard et al., 2002; Nakaoka et al., 2005). A direct approach is to imitate human hand motions from videos and retarget them to robotic embodiment (Sivakumar et al., 2022; Qiu et al., 2025; Luo et al., 2025). Recent advances in large-scale video datasets (Grauman et al., 2022; Hoque et al., 2025) and automated hand pose estimation frameworks (Pavlakos et al., 2024; Zhang et al., 2025) further enable this approach to scale. However, the embodiment gap—differences in physical properties, geometries, and kinematics between human and robot hands—remains a fundamental bottleneck, resulting in unstable grasps or unreachable poses.

To mitigate this gap, recent work optimizes neural control policies via RL with dual objectives: imitating human behaviors and achieving target object trajectories from videos (Chen et al., 2024a; Liu et al., 2025; Li et al., 2025). The imitation term guides exploration toward human-like behaviors, while the object-centric term ensures task success. These policies are trained in simulated environments reconstructed from videos before real-world deployment. However, this approach requires accurate ground-truth object poses, typically available only in motion-capture datasets (Fan et al., 2023; Banerjee et al., 2024; Zhan et al., 2024), severely limiting scalability for monocular RGB videos without pose labels.

Motivated by these limitations, we introduce DexMan, an automated framework that translates human visual demonstrations into bimanual dexterous manipulation skills for humanoid robots in simulation. Given only a third-person monocular RGB video of a human manipulating rigid objects—without camera calibration, ground-truth depths, or 3D object assets—DexMan reconstructs the 3D scene, recovers hand and object motions, and trains a residual RL policy that reproduces target object trajectories while being guided by human hand motion and physical contact priors. Unlike prior work that relies on motion-capture data or controls simplified floating hands, DexMan directly controls a full humanoid robot from raw videos, making it the first framework to derive feasible multi-arm, multi-fingered robot skills directly from monocular RGB inputs. Furthermore, DexMan can generate skills from synthetic video data (Brooks et al., 2024), avoiding the need for manual data collection and enabling the curation of large-scale datasets for training generalist manipulation policies (Liu et al., 2025).

We present several technical innovations to address key challenges in our video-to-robot skill acquisition pipeline. First, recovering 3D object motion demands accurate pose estimation, yet state-of-the-art analysis-by-synthesis methods (Wen et al., 2024; Lee et al., 2025) struggle with textureless objects. We leverage 3D point trajectories (Karaev et al., 2024; Xiao et al., 2025) as supplementary motion cues to improve pose accuracy. Second, reconstructed 3D objects often wobble or topple in simulation due to low-quality meshes; we propose a sampling-based method to identify stable configurations close to the original poses. Finally, training RL policies for bimanual dexterous manipulation remains highly challenging. We introduce contact-based rewards that guide the robot toward firm grasps, which are essential for reliable manipulation.

We evaluate DexMan across three challenging settings. First, on the motion-capture dataset TACO (Liu et al., 2024b), our pose estimation pipeline achieves consistently lower errors than all baselines, establishing a reliable foundation for downstream skill learning. Second, on the OakInk-v2 benchmark (Zhan et al., 2024), our residual RL policy outperforms the previous state of the art by an absolute margin of 19% in success rate (Li et al., 2025), despite controlling a full humanoid robot with two dexterous hands rather than simplified floating hands. Finally, we demonstrate the first end-to-end video-to-robot skill acquisition pipeline from uncalibrated monocular RGB inputs: without ground-truth hand–object annotations, DexMan successfully recovers 27.4% of skills from TACO videos and 37.0% from Veo3-generated synthetic videos (Google DeepMind, 2025), showcasing its potential to scale beyond motion-capture datasets toward generalizable dexterous manipulation.

## 2 RELATED WORK

**6D Object Pose Estimation** Recent research in 6D pose estimation has shifted towards generalizable approaches. FoundationPose (Wen et al., 2024) achieves state-of-the-art performance via unified estimation and tracking with neural implicit representation, while GigaPose (Nguyen et al., 2024) and MegaPose (Labbé et al., 2022) employ coarse-to-fine refinement. Any6D (Lee et al., 2025) provides single-frame estimation from a single RGB-D image, and SpatialTracker (Xiao et al., 2025)

enables robust temporal tracking through dense correspondence. We build upon FoundationPose and SpatialTracker for object pose estimation in our framework.

**Controllers for dexterous manipulation**   Three main approaches tackle multi-fingered manipulation: (1) Trajectory Optimization (Mordatch et al., 2012; Hwangbo et al., 2018; Pang & Tedrake, 2021; Pang et al., 2023; Jin, 2024; Liu et al., 2024a) uses known dynamics but struggles with contact modeling; (2) Imitation Learning (Arunachalam et al., 2022; Li et al., 2023; Qiu et al., 2025; Chen et al., 2022b; Wu et al., 2023; Lin et al., 2024; Wang et al., 2024) from teleoperation or planning demonstrations faces scalability issues due to labor-intensive data collection; (3) Reinforcement Learning (Rajeswaran et al., 2017; Akkaya et al., 2019; Christen et al., 2022; Chen et al., 2023) with curriculum learning (Xu et al., 2023; Yin et al., 2025) and BC-RL (Rajeswaran et al., 2017) addresses sample inefficiency in high-dimensional spaces.

**Learning manipulation from videos**   Transferring human videos to robot skills addresses data scarcity (Kareer et al., 2024). Methods detect 3D hand-object motions from video, then train RL policies for: single-arm jaw-gripper (Patel et al., 2022), single-arm dexterous (Mandikal & Grauman, 2022; Qin et al., 2022a;b; Sivakumar et al., 2022; Ye et al., 2023; Zhao et al., 2024; Chen et al., 2024b; Singh et al., 2024; Chen et al., 2025; Lum et al., 2025), dual-arm jaw-gripper (Zhou et al., 2025), and dual-arm dexterous manipulation (Li et al., 2024). While OKAMI (Li et al., 2024) requires RGB-D, DexMan derives bimanual dexterous actions directly from RGB videos, being the first to handle both real and synthetic inputs.

**Learning dexterous manipulation from reference human–object poses**   A recent line of work improves RL efficiency by leveraging human motion priors (Gupta et al., 2016; Garcia-Hernando et al., 2020; Zhou et al., 2024; Chen et al., 2024a; Luo et al., 2024; Li et al., 2025; Liu et al., 2025; Mandi et al., 2025). To address the high-dimensional action space of multi-fingered hands, these approaches often decouple tasks into a pre-grasp and a manipulation stage, or adopt residual policy learning to refine a base controller (e.g., PGDM(Dasari et al., 2023), DexTrack (Liu et al., 2025), OmniGrasp (Luo et al., 2024), MANIPTRANS (Li et al., 2025), DexMachina (Mandi et al., 2025)).

However, these methods typically assume access to motion-capture data. When reference motions are noisy or physically implausible, as in single-view video estimates, strict imitation can push policies toward infeasible behaviors, while relaxed imitation may lead to a local minimum where the policy avoids contact to minimize penalties. Existing contact-reward formulations are insufficient to guide the exploration when successful trajectory often deviates significantly from the provided reference motion. MANIPTRANS (Li et al., 2025), for instance, rewards proximity between hand keypoints and the nearest object surface; this encourages contact but does not differentiate task-relevant grasp regions from arbitrary surfaces. As a result, the policy often exploits trivial contacts rather than exploring the meaningful interactions needed to accomplish the task. DexMachina (Mandi et al., 2025) instead rewards reaching prerecorded world-space contact coordinates, but it is highly sensitive to object pose variation and encourages simple positional imitation. Both strategies may yield suboptimal solutions, such as touching the nearest surface or aligning with fixed spatial points. These limitations highlight the need for a structured, object-centric correspondence in reward design so that agents acquire generalizable contact skills rather than simply reaching predetermined positions.

## 3 METHOD

We propose DexMan, an automated framework that converts monocular RGB videos of human manipulation into bimanual dexterous robot skills in simulation. The pipeline comprises four stages: (1) reconstructing 3D objects from the input video, (2) recovering human hand and object motions, (3) building a stable interactive 3D scene in simulation and retargeting human motions to a humanoid robot embodiment, and (4) training a residual RL policy to reproduce target object trajectories under the guidance of human motion and contact priors. A key component is a novel contact reward that promotes firm, stable grasps and accelerates RL training. An overview of DexMan is shown in Fig. 2.

### 3.1 3D OBJECT RECONSTRUCTION

The core idea of DexMan is to reconstruct the necessary 3D scene from a human video and train a control policy within this environment to perform the same manipulation task as the human demonstrator. We define task success as the policy reproducing the 3D motion of the target object

Figure 2: **Overview of DexMan**. DexMan is a framework for acquiring robot skills from human videos. **Top:** From monocular input, DexMan reconstructs object meshes, estimates depth, and recovers 3D hand–object motions, then retargets these to a full humanoid robot in simulation (Makoviychuk et al., 2021) rather than floating hands. **Bottom:** A residual RL policy refines retargeted motions to reproduce object trajectories, guided by human motion and contact priors. DexMan introduces a contact reward that encourages stable grasps for effective RL training, enabling the robot to complete demonstrated manipulation tasks.

observed in the input video, while target objects refer to those manipulated by the demonstrators. To enable this, DexMan first identifies the target objects, tracks their motion trajectories, and constructs an interactive 3D scene for them. In our experiments, target objects are manually identified, though automated alternatives such as vision–language models with strong visual reasoning (Comanici et al., 2025) could also be adopted.

Building an interactive 3D scene requires accurate object assets, which monocular RGB videos do not provide. To recover them, DexMan employs an image-to-3D reconstruction pipeline. From the initial video frame, SAM2 (Ravi et al., 2024) segments the object mask, an image patch is cropped around the object, and Trellis (Xiang et al., 2024) generates rigid 3D meshes from the cropped image. We only consider reconstruction of target objects, ignoring those irrelevant to the manipulation task.

Notably, these meshes lack correct scale and pose, both of which are essential for reproducing the video scene in simulation. Accurate 3D hand and object poses are also needed to provide action priors and reward signals for policy training. In the following sections, we describe how DexMan rescales object meshes, and recovers hand and object 3D poses.

## 3.2 HAND AND OBJECT POSE ESTIMATION

Our key insight is to exploit estimated video depth maps to rescale object meshes and recover accurate object poses. This depth information captures the relative size of objects with respect to the human hand and enables reliable 3D pose estimation. Building on this idea, we design a staged pipeline that estimates video depths and hand poses, rescales object meshes, and refines object pose predictions.

**Depth estimation** DexMan leverages VGGT (Wang et al., 2025), an image-to-3D foundation model that predicts depth from sequences of frames. To process long videos, DexMan splits them into overlapping chunks, applying VGGT to each chunk independently. While VGGT generally produces temporally consistent outputs, with depth values remaining on similar scales across consecutive

frames, inconsistencies may still arise. In such cases, we align depth values within each object region across the entire video. Additional details are provided in Appendix B.1.

**Hand pose estimation** DexMan adopts HaMeR (Pavlakos et al., 2024), a hand mesh recovery framework that outputs metrically accurate MANO hand meshes (Romero et al., 2022). Since VGGT predicts only relative depth, we compute a global scale factor in the first frame by aligning the width of the MANO mesh with that of the VGGT point cloud. This factor is then applied uniformly to rescale all VGGT depth maps. Additional details are provided in Appendix B.2.

**Object pose estimation** DexMan estimates object scale following Any6D (Lee et al., 2025): we sample multiple candidate scales, use FoundationPose (Wen et al., 2024) to estimate poses for each, and select the scale associated with the highest-scoring pose. While FoundationPose generally produces strong results, its performance can degrade under fast object motion or heavy occlusion, leading to inconsistencies. To address this, we track 3D point trajectories with SpatialTracker (Xiao et al., 2025) and compute the rigid transformation between consecutive frames. We apply this transformation to the previous pose to obtain an updated initial pose, which is then provided to FoundationPose for refinement. This yields smoother and more temporally consistent estimates than using the unadjusted previous pose. We refer to Appendix B.3 for more details.

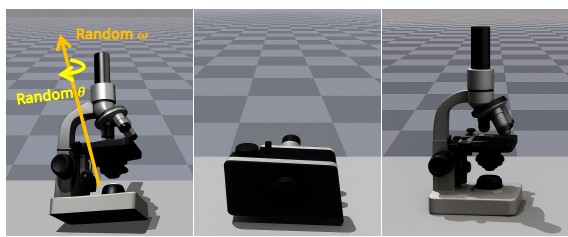

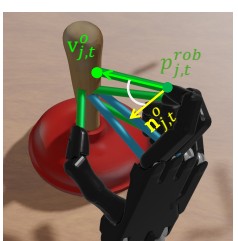

Figure 3: **Sampling stable object configuration**. DexMan perturbs object poses with random axes and angles, simulates each configuration, and selects the stable one closest to the original pose for placement in simulation.

Figure 4: **Contact reward.** The attraction term pulls robot hand keypoints $\mathbf{p}_{j,t}^{\text{robot}}$ toward human-contacted object vertices $\mathbf{v}_{j,t}$ and aligns the keypoint–vertex vector with the surface normal $\mathbf{n}_{j,t}^{\text{robot}}$, ensuring within-grasp contacts.

### 3.3 MOTION RETARGETING FROM HUMANS TO HUMANOIDS

The goal of this work is to learn neural policies in simulation that enable a humanoid robot to replicate human manipulation tasks, guided by human motion. This requires placing reconstructed 3D objects, a table and the robot in simulation, and retargeting human motions to the robot's embodiment.

**Placing a humanoid robot and objects in simulation** Unlike prior work that considers manipulation with floating hands (Li et al., 2025; Mandi et al., 2025), DexMan addresses a more realistic and challenging setting: controlling a humanoid robot to manipulate objects. We set up the simulation by placing a humanoid robot at the origin, oriented along the simulator's $y$-axis. For dexterous manipulation, each arm is equipped with a dexterous hand.

To enable robot manipulation of reconstructed objects in simulation, DexMan aligns the human body pose from the video's camera frame with the robot's simulator frame. Since camera extrinsics are unavailable, this alignment is approximated through three spatial transformations. First, the system rotates the scene so that the camera's gravity direction—approximated from the table surface normal via singular value decomposition of the table's point cloud—matches the simulator's gravity. Next, it rotates the human's hip-to-hip vector to align with the simulator's $x$-axis, which naturally orients the human body to face forward. Finally, it translates all entities to place them within the robot's workspace. Further details are provided in Appendix B.4.

Despite their high visual fidelity, reconstructed 3D objects are often physically incompatible (Guo et al., 2024). Direct placement can yield unstable configurations, causing objects to wobble or topple under gravity. To ensure stability, DexMan refines each object's initial pose through simulation. Specifically, it perturbs the original pose with 20 random rotations (up to $\pm 45°$ around sampled

axes), simulates each configuration for 20 steps, and selects the stable pose with minimal rotational deviation from the original. This reduces physical instability in the environment and makes RL policy training more effective.

**Human-to-robot motion retargeting** To leverage human motion priors, DexMan retargets the demonstrator's wrist and finger motions to the robot embodiment for bimanual dexterous manipulation. Solving this problem jointly is inefficient due to the high degrees of freedom of humanoid robots. Instead, DexMan adopts a staged strategy that handles wrist and finger retargeting separately. For wrist retargeting, DexMan employs an inverse kinematics (IK) solver (Genesis, 2024) to compute the required joint angles, keeping the lower body fixed since tabletop tasks do not involve lower-body motion. For finger retargeting, DexMan trains a neural IK solver via supervised learning. The solver is a 5-layer MLP that takes as input the 3D positions of the five fingertips of either hand and outputs the corresponding finger joint angles.

### 3.4 REINFORCEMENT LEARNING WITH NOISY MOTION PRIORS

While motion retargeting can translate human motions to the robot's embodiment, it does not address physical inconsistency or noise of estimated human motions, leading to failed task execution. To overcome this, we perform RL to transfer noisy retargeted motions into feasible robot skills.

**Contact-Prior Attraction Reward** Our approach leverages contact prior to provide robust guidance for dexterous manipulation when learning from noisy reference motion. The core idea is to shift the learning objective away from imitating ineffective hand trajectories towards fulfilling object-centric contact goals. The reward works by establishing correspondence between designated hand keypoints and their intended near-contact regions on the object's surface.

By focusing on this local relationship, our method directly addresses the fundamental dilemma introduced by noisy motion data. Instead of being forced to imitate a physically implausible global trajectory, the policy is guided by a more reliable prior–the intended spatial relationship at the point of contact. This provides a clear and stable learning signal that encourages the agent to make meaningful contact, steering it away from the local minimum contact avoidance.

Furthermore, this formulation overcomes the key limitations of prior contact rewards discussed in the related work. Unlike rewards that simply encourage touching the nearest point on an object's surface (Li et al., 2025), our approach provides a structured correspondence, guiding exploration efficiently toward functionally critical contacts instead of arbitrary ones. In contrast to rewards based on static, world-space coordinates (Mandi et al., 2025), its object-centric nature makes the learned policy inherently robust to variations in the object's pose.

**Offline contact-prior extraction** Let $\mathcal{J}$ be the set of hand keypoints such as fingertips or palm keypoints, $\mathcal{O}$ be the set of manipulated objects. For keypoint $j$ at timestep $t$, we denote $\mathbf{p}_{j,t}^{\text{hum}} \in \mathbb{R}^3$ as the human hand keypoint position estimated from the video, $\mathbf{p}_{j,t}^{\text{rob}} \in \mathbb{R}^3$ and $\mathbf{n}_{j,t}^{\text{rob}} \in \mathbb{R}^3$ as the robot hand keypoint position and its outward surface normal during training. Our goal is to identify keypoint–vertex pairs that correspond to hand–object contacts. For each hand keypoint $j \in \mathcal{J}$ at time $t$, we find the closest mesh vertex on every object $o \in \mathcal{O}$: $\mathbf{v}_{j,t}^{o} \in \mathbb{R}^3 = \arg\min_{\mathbf{v} \in \mathcal{V}_o} \left\| \mathbf{p}_{j,t}^{\text{hum}} - \mathbf{v} \right\|_2$, where $\mathcal{V}_o$ denotes the set of mesh vertices on object $o$. We then construct a filtered set of contact candidates $\mathcal{P}(t)$, keeping only those pairs where the keypoint–vertex distance is below a threshold: $\mathcal{P}(t) = \{(j,o) | \forall j \in \mathcal{J}, \forall o \in \mathcal{O}, \|\mathbf{p}_{j,t}^{\text{hum}} - \mathbf{v}_{j,t}^{o}\|_2 \leq \tau_j\}$ with $\tau_j$ as the threshold at keypoint $j$.

**Online attraction rewards** The contact reward includes two terms: (1) an attraction term that pull the robot keypoints toward the human-contacted object vertices, and (2) a directional alignment term that prevents unrealistic back-of-hand contacts. Formally, let $\widehat{\mathbf{d}}_{j,t}^{o} = \frac{\mathbf{v}_{j,t}^{o} - \mathbf{p}_{j,t}^{\text{rob}}}{\left\| \mathbf{v}_{j,t}^{o} - \mathbf{p}_{j,t}^{\text{rob}} \right\|}$ be the directional vector from a robot keypoint to the associated object vertex, and $\mathbf{n}_{j,t}^{\text{rob}}$ be its surface normal. We formulate the contact reward as follows:

$$R_{\text{contact}}(t) = \frac{1}{|\mathcal{O}|} \sum_{(j,o) \in \mathcal{P}(t)} w_c^j (1 + \gamma_c^j \langle \mathbf{n}_{j,t}^{\text{rob}}, \widehat{\mathbf{d}}_{j,t}^{o} \rangle) e^{-\lambda_c \left\| \mathbf{p}_{j,t}^{\text{rob}} - \mathbf{v}_{j,t}^{o} \right\|_2^2} (w_{c_1} + w_{c_2} \mathbf{1}_{\text{lifted}}^{o}) \quad (1)$$

where $\mathbf{1}_{\text{lifted}}^{o}$ denotes an indicator function representing if object $o$ is lifted.

**Object-following rewards** We adopt the object-following reward to enforce reproduction of target object trajectories. Let $\Delta_{\text{pos}}^{o}$ and $\Delta_{\text{rot}}^{o}$ denote the positional and rotational differences between the

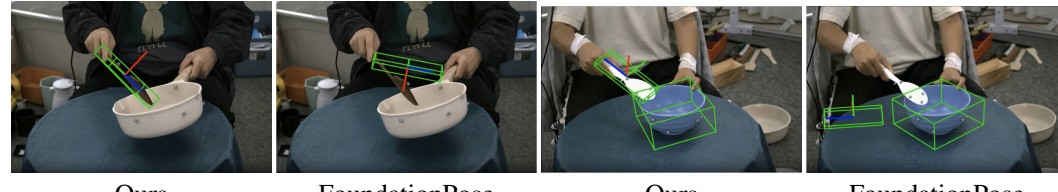

| Ours | FoundationPose | Ours | FoundationPose |

Figure 5: **Visual comparison of object pose estimation**. We show the estimated object pose with an oriented 3D bounding box, along with three coordinate axes. Our method incorporates additional motion cues–3D point trajectories, producing more stable and accurate pose estimation than FoundationPose's outputs (Wen et al., 2024).

current and reference poses of object $o$ at timestep $t$. The reward is formulated as:

$$R_{\text{obj}}(t) = \frac{1}{|\mathcal{O}|} \sum_{o \in \mathcal{O}} (w_o^{\text{pos}} e^{-\lambda_o^{\text{pos}} \Delta_{\text{pos}}^o} + w_o^{\text{rot}} e^{-\lambda_o^{\text{rot}} \Delta_{\text{rot}}^o})(w_{o_1} + w_{o_2} \mathbf{1}_{\text{lifted}}^o) \quad (2)$$

**Imitation rewards**  We use the imitation reward to enforce imitation of human hand motions. Let $\Delta_{\text{pos}}^{\text{hand}}, \Delta_{\text{rot}}^{\text{hand}}, \Delta_{\text{jnt}}^{\text{hand}}$ be the wrist positional, wrist rotational and per-joint finger positional difference. This reward is: $R_{\text{imit}}(t) = w_i^{\text{pos}} e^{-\lambda_i^{\text{pos}} \Delta_{\text{pos}}^{\text{hand}}} + w_i^{\text{rot}} e^{-\lambda_i^{\text{rot}} \Delta_{\text{pos}}^{\text{hand}}} + w_i^{\text{jnt}} \sum_{\text{jnt}} e^{-\lambda_i^{\text{jnt}} \Delta_{\text{jnt}}^{\text{hand}}}$

Finally, the total reward combines these three rewards: $R(t) = R_{\text{contact}}(t) + R_{\text{obj}}(t) + R_{\text{imit}}(t)$.

**Policy and Control**  We train a residual RL policy to refine retargeted human motions. At each time step $t$, the policy observes the robot's proprioception and object states, and predicts residual corrections $\{\Delta x_t^h, \ \Delta \omega_t^h, \ \Delta \theta_t^h\}$ to wrist position, wrist orientation, and hand joint angles. These residuals are converted into executable robot joint angles using the same set of IK solvers as in motion retargeting, and executed via a PD controller. Further details are provided in Appendix B.5.

## 4 EXPERIMENTS

| | Accuracy | | Robustness | |
|---|---|---|---|---|
| | ADD-S ↑ | VSD ↑ | Failure Rate ↓ | Temp. Stability ↑ |
| FoundationPose | 0.49 | 0.70 | 0.14 | 0.70 |
| SpatialTracker | 0.55 | 0.56 | 0.13 | **0.79** |
| DexMan (ours) | **0.57** | **0.82** | **0.01** | 0.76 |

Table 1: **Pose estimation performance on TACO.** DexMan outperforms both baselines in accuracy metrics (ADD-S, VSD) and failure rate reduction.

### 4.1 OBJECT POSE ESTIMATION

We evaluate our object pose estimation pipeline on TACO (Liu et al., 2024b), a motion-capture dataset containing 244 bimanual manipulation sequences with ground-truth hand-object poses. We compare against two state-of-the-art baselines: (1) FoundationPose (Wen et al., 2024), which directly predicts 6DoF object poses from RGB-D images; and (2) SpatialTracker (Xiao et al., 2025), which tracks 3D point motions to derive rigid transformations. Our method integrates both approaches by regularizing pose estimation with point trajectories.

We adopt the following evaluation metrics: ADD-S (Xiang et al., 2018), VSD(Hodaň et al., 2018), failure rate Kristan et al. (2023), and temporal stability Sturm et al. (2012) (see Appendix B.6 for details). As presented in Table 1, our method consistently outperforms baselines with gains of 0.08, 0.12, 0.13, and 0.06 respectively (Table 1). Figure 5 further demonstrates superior 3D motion estimates in challenging scenarios.

|  | Success Rate ↑ | $E_r \downarrow$ | $E_t \downarrow$ |
|---|---|---|---|
| MANIPTRANS | 25.3 | 0.180 | 0.00646 |
| DexMan (ours) | **44.3** | 0.178 | 0.00688 |

Table 2: **Bimanual dexterous manipulation on OakInk-v2**. Both MANIPTRANS and DexMan use ground-truth object assets, hand and object motion annotations provided by OakInk-v2 motion-capture dataset. DexMan achieves significantly higher success rate than the baseline.

## 4.2 RESIDUAL RL POLICY

We evaluate our residual RL policy on OakInk-v2 (Zhan et al., 2024), a challenging bimanual dexterous manipulation benchmark with ground-truth pose annotations. Following Li et al. (2025), we test on 77 motion sequences from the validation set featuring humans bimanually manipulating rigid objects. Our simulation setup uses the Unitree G1 humanoid robot (Robotics, 2024) equipped with Shadow Dexterous Hands (Plappert et al., 2018) on each arm, operating under position-controlled joint angles. We compare against MANIPTRANS (Li et al., 2025), a state-of-the-art approach for transferring human bimanual skills that focuses on floating robotic hands, while our method controls a full humanoid—offering a more realistic yet challenging setting. Both methods are trained with PPO on NVIDIA Isaac Gym for 500 RL iterations, collecting rollouts of 32 steps from 2,048 parallel environments per update.

We adopt two standard metrics: success rate and average object rotation/translation errors ($E_r$ and $E_t$), computed only for successful rollouts following MANIPTRANS. However, we extend the evaluation protocol to consider entire episodes rather than just intermediate frames as in MANIPTRANS. Each sequence is evaluated over 10 independent rollouts, with success requiring rotational error below 0.5 radians and positional error below 3cm at every time step (using MANIPTRANS thresholds). As shown in Table 2, our method achieves 19% higher success rate than MANIPTRANS, which we attribute to our contact-based reward promoting stable grasps for more reliable task execution.[1]

| **TACO Dataset** | Success Rate (%) ↑ | $E_r$(rad) ↓ | $E_t$ (m) ↓ | Traj→Mask IoU (%) ↑ |
|---|---|---|---|---|
| DexMan (ours) | **27.4** | 0.314 | 0.016 | 49.0 |
| w/o task reward | 18.4 | 0.330 | 0.022 | 42.8 |
| w/o contact reward | 7.2 | 0.182 | 0.018 | 61.9 |
| **Synthetic Videos** | Success Rate (%) ↑ | $E_r$(rad) ↓ | $E_t$ (m) ↓ | Traj→Mask IoU (%) ↑ |
| DexMan (ours) | **39.0** | 0.248 | 0.017 | 44.7 |
| w/o task reward | 34.6 | 0.288 | 0.023 | 41.3 |
| w/o contact reward | 7.8 | 0.165 | 0.013 | 59.9 |

Table 3: **Video-to-robot skill acquisition on TACO and synthetic videos**. For TACO, we do not use any ground-truth object asset, hand and object motion annotation provided. For synthetic videos, we generate videos using Veo3 (Google DeepMind, 2025) without ground-truth 3D assets or motion annotations. The proposed contact reward enables DexMan to recover physically plausible robot skills directly from monocular RGB videos.

## 4.3 SKILL ACQUISITION FROM REAL AND SYNTHETIC VIDEOS

We evaluate the full video-to-robot skill acquisition pipeline on real videos from TACO and synthetic videos. We select 50 out of 244 motion sequences from the TACO prereleased dataset, and generate 50 videos of humans performing bimanual manipulation tasks with Veo3 (Google DeepMind, 2025). We use the same training setup as in Section 4.2, but extend training to 1,000 policy updates to account for the increased task complexity.

We adopt the following metrics: (1) success rate, (2) $E_r$ and $E_t$ as defined previously, and (3) intersection-over-union (IoU) between detected object masks and those rendered from simulated

---

[1]See discussion at https://github.com/ManipTrans/ManipTrans/issues/18

object motions. Since ground-truth object motion is unavailable in this setup, IoU serves as a proxy for assessing the realism of manipulated object trajectories. Note that $E_r$, $E_t$, and IoU are computed only for successful rollouts. Consequently, when success rate is low, these metrics may be biased toward easier successful cases.

We do not compare against other baselines as, to the best of our knowledge, DexMan is the first framework capable of transferring monocular RGB human videos into bimanual dexterous robot policies. As shown in Tables 3, DexMan achieves success rates of 27.4% on TACO and 39.0% on synthetic videos, without relying on ground-truth hand–object motion annotations. DexMan closely reproduces the observed object motions, reaching IoU scores of 49.0% on TACO and 44.7% on Veo3-generated videos.

**Ablation study** We analyze the contribution of each reward component in Table 3. Though task rewards are also important—removing them reduces success rates by 9.0% on TACO and 4.4% on Veo3 videos—our results show that contact rewards are even more critical. Without them, success rates collapse from 27.4% to 7.2% on TACO and from 39.0% to 7.8% on Veo3 videos, highlighting that contact guidance is the dominant factor for recovering physically plausible robot skills directly from monocular videos.

Finally, we assess the effect of sampling stable object configurations. As shown in Table 4, our sampling strategy achieves 98% of stable initializations across 50 TACO and 50 Veo3-generated videos, compared to 86% without sampling. A configuration is considered stable if it settles within the 0.75 rad initialization threshold. Importantly, even the 86% "stable" unsampled configurations exhibit residual pose errors. Although these errors remain below the failure threshold, they can accumulate over time and degrade RL training performance. This highlights the value of our sampling approach.

| Sampling | Stable configurations (%) |
|:---:|:---:|
| ✓ | 98% |
| ✗ | 86% |

Table 4: **Effect of object configuration sampling**. Our sampling strategy significantly increases the percentage of stable initial configurations.

## 5 LIMITATIONS

Our framework has been developed and evaluated exclusively in simulation environments without deployment on physical robots, leaving a substantial sim-to-real gap to be addressed. Additionally, the assumptions about the scene are restrictive, as we only handle single-human demonstrations with rigid tabletop objects. Real-world scenarios, however, often involve deformable or articulated objects with complex joint configurations, which our current system cannot accommodate.

The system prioritizes task completion over motion naturalness, resulting in robot trajectories that deviate from human movements. This issue is compounded by the robot's constrained action space, which limits motion replication capabilities and suggests the need for mobile manipulation platforms. Furthermore, our end-effector control parameterization overlooks full arm posture, which is crucial for collision avoidance and effective multi-arm coordination in complex manipulation scenarios.

Our sequential pipeline estimates hand and object poses separately before extracting contact priors, which may compromise the consistency of contact reasoning—a fundamental requirement for manipulation tasks. This decoupled approach could introduce errors that propagate through the system. A more integrated approach that explicitly reasons about contact points during pose estimation could provide more reliable priors for policy learning and improve overall system performance.

## 6 SUMMARY

We presented DexMan, a novel framework that converts monocular human videos into bimanual dexterous robot skills without requiring calibration, depth information, 3D assets, or motion labels. By combining object reconstruction, hand-object retargeting, and contact-centric residual policy refinement, our approach achieves state-of-the-art performance on TACO and OakInk-v2 benchmarks. Despite current limitations in sim-to-real transfer and motion naturalness, DexMan demonstrates a scalable pathway for robot skill acquisition from human demonstrations, paving the way for future work in real-world deployment and more complex manipulation scenarios.

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

APPENDIX

# A DECLARATION OF LLM USAGE

We used large language models (LLMs) to assist with writing refinement (e.g., grammar, spelling, word choice), to enhance image resolution in Fig. 2 and Fig. 3, and to support code implementations.

# B METHOD DETAILS

## B.1 SPATIO-TEMPORALLY CONSISTENT DEPTH ESTIMATION

DexMan leverages VGGT (Wang et al., 2025), an image-to-3D foundation model that predicts depth from sequences of frames. To process long videos, DexMan splits them into overlapping chunks, applying VGGT to each chunk independently. However, this independent processing leads to scale inconsistencies across chunks, which in turn make object pose estimation unreliable.

Ensuring temporal consistency in estimated depths remains a significant challenge. For a given video frame, depth maps predicted from consecutive chunks often differ in complex, non-linear ways. This makes it infeasible to align them with a single global transformation. To address this, we propose an *object-centric* temporal alignment strategy, which computes a local linear mapping to align depth values within each object's image region independently. Formally, let $D_{k-1}^o$ and $D_k^o$ denote the sets of depth values for object $o$ in a video frame, obtained from an overlapping frame of the $(k-1)$-th and $k$-th video chunks. We solve for affine parameters $\alpha_k, \beta_k$ by minimizing: $\min_{\alpha_k, \beta_k} ||\alpha_k D_k^o + \beta_k - D_{k-1}^o||_2$. The resulting affine mapping is then applied to all pixels of object $o$ across the frames in chunk $k$. We restrict this alignment procedure to human hands and target objects, which are tracked and segmented using SAM2 (Ravi et al., 2024). By iteratively propagating object-centric depth alignment across all video chunks, we enforce spatio-temporal consistency of depth values while preserving relative object sizes with respect to the human hand.

## B.2 RESCALE VGGT DEPTH MAPS WITH MANO'S OUTPUT

We first obtain metrically accurate right-hand meshes and a corresponding 2D mask in the first frame using (Goel et al., 2023). The mask is then projected using depth maps predicted by VGGT. To align

the depth maps with the metric scale of HaMeR, we compute a scale factor as

$$\text{scale} = \frac{x_{\max}^{\text{HaMeR}} - x_{\min}^{\text{HaMeR}}}{x_{\max}^{\text{VGGT}} - x_{\min}^{\text{VGGT}}},$$

where $x_{\max}^{\text{HaMeR}}$ and $x_{\min}^{\text{HaMeR}}$ denote the horizontal extents of the HaMeR mesh, and $x_{\max}^{\text{VGGT}}$ and $x_{\min}^{\text{VGGT}}$ denote those of the unprojected VGGT point cloud. The computed scale factor is then applied uniformly to all frames of the VGGT depth maps.

### B.3 RIGID TRANSFORMATIONS OF POINT MOTIONS

We first obtain corresponded 3D pixel positions of each object by tracking point trajectories across frames using SpatialTracker (Xiao et al., 2025).

Formally, let $P_{k-1}^o$ and $P_k^o$ be the sets of corresponded 3D pixel positions of object $o$ in video frame $k-1$ and $k$. DexMan calculates their rigid transformations $\Delta T_{k-1}^k \in SE(3)$ with Kabsh algorithm (Arun et al., 1987): $\min_{\Delta T} \|\Delta T \bar{P}_{k-1}^o - \bar{P}_k^o\|_F^2$, where $\bar{P}_k^o$ denotes the homogeneous coordinates of $\bar{P}_k^o$.

### B.4 ALIGNING THE HUMAN BODY POSE IN VIDEO WITH THE ROBOT POSE IN SIMULATION

To enable the robot to manipulate reconstructed objects in simulation, the objects, hand keypoints, and a table must be placed in configurations consistent with the robot's embodiment. DexMan estimates a spatial transformation that aligns the human body pose in the video with the robot's pose in the simulator. This same transformation is then applied to transfer recovered object poses, 3D hand keypoints, and the table point cloud–derived directly from estimated depth maps–from the camera coordinate frame into the simulator's coordinate frame. In this work, DexMan computes three consecutive transformations: a rotation that aligns with the simulator's gravity direction, a rotation that aligns with the robot's facing direction, and a translation that places all reconstructed entities near the robot.

In simulation, gravity is applied along $z$-axis, but our estimated hand–object poses are expressed in the camera coordinate frame of the video, whose axes are misaligned with gravity. To correct this, DexMan identifies the gravity direction in the camera frame, computes a rotation matrix to align it with the simulation's gravity axis, and applies this transform to all object poses, 3D hand keypoints and the table point cloud. Since we focus on tabletop manipulation, the gravity vector is approximated by the surface normal of the table. Specifically, DexMan detects the table bounding box with OWLv2 (Minderer et al., 2023), segments it using SAM2 (Ravi et al., 2024), lifts the segmented pixels into 3D with estimated depth maps, and derives the surface normal as the principal axis with the smallest eigenvalue[2] from singular value decomposition of the resulting point cloud. The initial video frame is used to align with gravity.

To align with the robot's facing direction, the human body pose in the video serves as an anchor to derive the required spatial transformation. DexMan first applies HUMAN4D (Goel et al., 2023) to detect 3D keypoints of the human's left- and right-hip ($p_{\text{l.hip}}^{\text{hum}}, p_{\text{r.hip}}^{\text{hum}}$), calculates their residual vector $p_{\text{r.hip}}^{\text{hum}} - p_{\text{l.hip}}^{\text{hum}}$, and finds a rotation matrix that orients the vector toward the simulator's $x$-axis.

The translation that moves reconstructed entities near the robot is derived from the positional residual between the human and robot bodies. DexMan detects the 3D human pelvis keypoint from the video, applies the previously computed rotations, and measures the offset between the transformed human pelvis and the robot's pelvis position. It defines the translation. If the reconstructed entities remain outside the robot's workspace, an additional translation along the simulator's $y$-axis is applied.

### B.5 RESIDUAL RL POLICY

We follow (Li et al., 2025) to set the ShadowHand friction coefficient to 4. Our robot is operated at a control frequency of 30 Hz. We train policies using Proximal Policy Optimization (PPO) implemented

---

[2]The table point cloud forms a 3D plane, whose principal component analysis yields two in-plane axes and one axis corresponding to the surface normal.

in RL_GAMES(Makoviichuk & Makoviychuk, 2021). The simulator and training configuration used in our Isaac Gym(Makoviychuk et al., 2021) enviroments are summarized in Table 5.

Table 5: Simulator and training configuration in Isaac Gym. ShadowHand friction is set to 4 following (Li et al., 2025).

| Parameter | Value |
|---|---|
| Algorithm | PPO |
| Policy Network | Actor-Critic (MLP: [256, 512, 128, 64], activation: ELU) |
| Learning Rate | $5 \times 10^{-4}$ (warmup schedule) |
| Discount Factor $\gamma$ | 0.99 |
| GAE Parameter $\tau$ | 0.95 |
| Horizon Length | 32 |
| Minibatch Size | 1024 |
| Mini-epochs | 5 |
| Entropy Coefficient | 0.0 |
| Critic Coefficient | 4 |
| Gradient Norm Clip | 1.0 |
| Clipping $\epsilon$ | 0.2 |
| Bounds Loss Coefficient | $1 \times 10^{-4}$ |
| dt | 1 / 30 |
| Substeps | 4 |
| ShadowHand Friction | 4.0 |
| Object Friction | 1.0 |
| Table Friction | 0.25 |

We use state-based reinforcement learning. The policy directly observes low-dimensional state features rather than raw visual inputs. Specifically, our observation space consists of (1) the robot's joint states, (2) the object pose, (3) the target pose, and (4) the current timestep. These quantities are summarized in Table 6.

Table 6: Observation space used for state-based RL training.

| Observation | Description |
|---|---|
| robot joints state | Joint positions and velocities of the humanoid |
| reference hand joints | Joint angles of the reference human hand motion |
| current wrist pose | 6D poses of both left and right wrists |
| reference wrist pose | 6D wrist pose from the reference motion |
| object pose | 6D pose of the manipulated object |
| target pose | 6D target pose of the manipulated object |

**Reward weights and thresholds.** Let $\mathcal{J}$ denote the set of hand keypoints consisting of (i) five fingertips and (ii) palm keypoints at the metacarpophalangeal (MCP) joints of the index, middle, ring, and little fingers. We use per-keypoint distance thresholds $\tau_j$ given by

$$\tau_j = \begin{cases} 0.03, & j \in \text{fingertips}, \\ 0.05, & j \in \text{palm (MCP) keypoints}. \end{cases}$$

Table 7: RL reward weights and hyperparameters.

| Contact reward | |
|---|---|
| $\gamma_c^j$ | $1 \;\; \forall j \in \mathcal{J}$ |
| $w_c^j$ | 0.5 (fingertips), 2.0 (palm keypoints) |
| $\lambda_c$ | 400 |
| $w_{c_1}$ | 1.0 |
| $w_{c_2}$ | 1.0 |
| **Object-following reward** | |
| $w_o^{\text{pos}}$ | 5 |
| $w_o^{\text{rot}}$ | 1 |
| $\lambda_o^{\text{pos}}$ | 80 |
| $\lambda_o^{\text{rot}}$ | 3 |
| $w_{o_1}$ | 0.1 |
| $w_{o_2}$ | 4 |
| **Imitation reward** | |
| $w_i^{\text{pos}}$ | 0.5 |
| $w_i^{\text{rot}}$ | 0.5 |
| $w_i^{\text{jnt}}$ | 0.5 |
| $\lambda_i^{\text{pos}}$ | 2 |
| $\lambda_i^{\text{rot}}$ | 20 |
| $\lambda_i^{\text{jnt}}$ | 20 |

### B.6 METRICS FOR POSE ESTIMATION

**Setup.** All metrics are computed in the world frame. Predicted poses $\{T_t\}_{t=1}^T$ (converted from a particular camera via its extrinsics) are first aligned to the ground-truth trajectory $\{T_t^\star\}_{t=1}^T$ using a *fixed* first-frame alignment computed once at $t=1$ and applied to every frame:

$$R_{\text{align}} = R_1^\star R_1^\top, \quad t_{\text{offset}} = t_1^\star - R_{\text{align}} t_1, \quad \tilde{T}_t = \begin{bmatrix} R_{\text{align}} R_t & R_{\text{align}} t_t + t_{\text{offset}} \\ \mathbf{0}^\top & 1 \end{bmatrix}.$$

We then score $\tilde{T}_t$ against $T_t^\star$.

**(1) ADD-S (Xiang et al., 2018).** Let $\mathcal{M} = \{x_i\}_{i=1}^N$ be model vertices in the object frame, and let $T = [R|t] \in SE(3)$. The symmetric ADD error is the average closest-point distance between the model transformed by the estimated and ground-truth poses:

$$d_{\text{ADD-S}}(\tilde{T}, T^\star) = \frac{1}{N} \sum_{i=1}^N \min_{j \in \{1, \dots, N\}} \|(Rx_i + t) - (R^\star x_j + t^\star)\|_2 \quad \text{(meters; lower is better).}$$

**(2) VSD (visible-surface consistency) (Hodaň et al., 2016; 2018).** Given an observed depth map $D \in \mathbb{R}^{H \times W}$ and a binary visible-mask $M \in \{0, 1\}^{H \times W}$, we project model points under $\tilde{T}$ to pixels $(u, v)$ with rendered depth $z$ and keep only pixels with $M(v, u) = 1$ and $D(v, u) > 0$. The per-frame VSD score is the fraction of visible pixels whose depth agrees within a tolerance $\delta$:

$$s_{\text{VSD}}(\tilde{T}; D, M) = \frac{1}{N_v} \sum_{(u,v) \in \Omega_{\text{vis}}} \mathbb{1}\!\left(|z - D(v, u)| < \delta\right), \qquad \delta = 0.02 \text{ m,}$$

where $\Omega_{\text{vis}}$ is the set of valid visible pixels and $N_v = |\Omega_{\text{vis}}|$. Higher is better. We report an AUC-style aggregate by sweeping a threshold on the per-frame score, $\tau \in [0.1, 0.5]$:

$$\text{AUC}_{\text{VSD}} = \frac{1}{\tau_{\max} - \tau_{\min}} \int_{\tau_{\min}}^{\tau_{\max}} \frac{1}{T} \sum_{t=1}^T \mathbb{1}\!\left(s_{\text{VSD}}^{(t)} \geq \tau\right) d\tau,$$

and additionally the success rate at a fixed operating point (e.g., $s_{\text{VSD}} > 0.3$).

**(3) Failure rate (VOTS-style robustness) Kristan et al. (2023).** After filtering frames with missing masks or invalid depth, we render a binary silhouette $\hat{M}_t$ from $\tilde{T}_t$ (via vertex projection with a 1-pixel cross-shaped dilation) and compute IoU with the observed mask $M_t$:

$$\text{IoU}(\tilde{T}_t) = \frac{|\hat{M}_t \cap M_t|}{|\hat{M}_t \cup M_t|}.$$

A frame is counted as a failure if it is invalid *or* $\text{IoU}(\tilde{T}_t) < \tau_{\text{IoU}}$ with $\tau_{\text{IoU}}{=}0.1$. The failure rate is

$$\text{FailRate} = \frac{1}{T} \sum_{t=1}^{T} \mathbb{1}\Big(\text{invalid}_t \text{ or } \text{IoU}(\tilde{T}_t) < 0.1\Big) \quad \text{(lower is better)}.$$

**(4) Temporal stability (RPE-based) Sturm et al. (2012).** Let $\Delta T_t^\star = (T_t^\star)^{-1} T_{t+1}^\star$ and $\Delta \tilde{T}_t = \tilde{T}_t^{-1} \tilde{T}_{t+1}$ be successive relative motions. Define translational and rotational discrepancies

$$e_t^{\text{trans}} = \big\| \Delta \tilde{t}_t - \Delta t_t^\star \big\|_2, \quad e_t^{\text{rot}} = \arccos\left(\frac{\text{tr}(\Delta \tilde{R}_t \Delta R_t^{\star\top}) - 1}{2}\right),$$

and map them to a $[0,1]$ smoothness score

$$s_{\text{stab}}(t) = \tfrac{1}{2} \exp\Big(-\frac{e_t^{\text{trans}}}{0.01}\Big) + \tfrac{1}{2} \exp\Big(-\frac{e_t^{\text{rot}}}{0.1}\Big),$$

where $0.01\,\text{m}$ and $0.1\,\text{rad}$ set the characteristic scales. We report the mean (and standard deviation) of $s_{\text{stab}}(t)$ over $t$ (higher is better).

## C    EXPERIMENTAL RESULTS

### C.1    IMPLAUSIBLE MOTION REFERENCES TO SUCCESSFUL POLICY

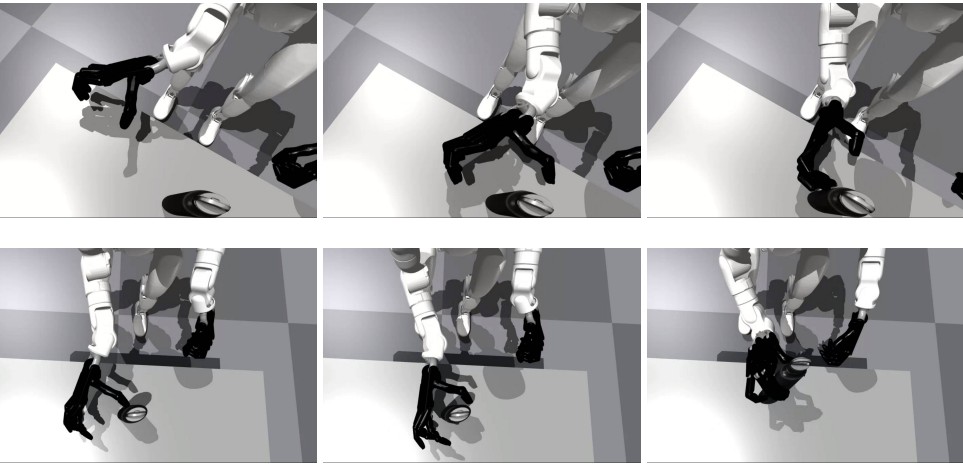

Figure 6: Failure (top row) and success (bottom row) cases for implausible bottle manipulation.

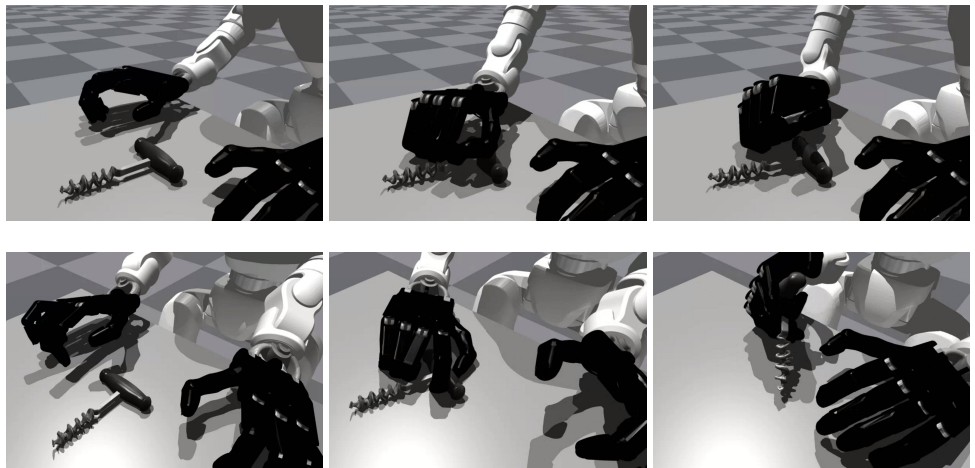

Figure 7: Failure (top row) and success (bottom row) cases for implausible cork manipulation.

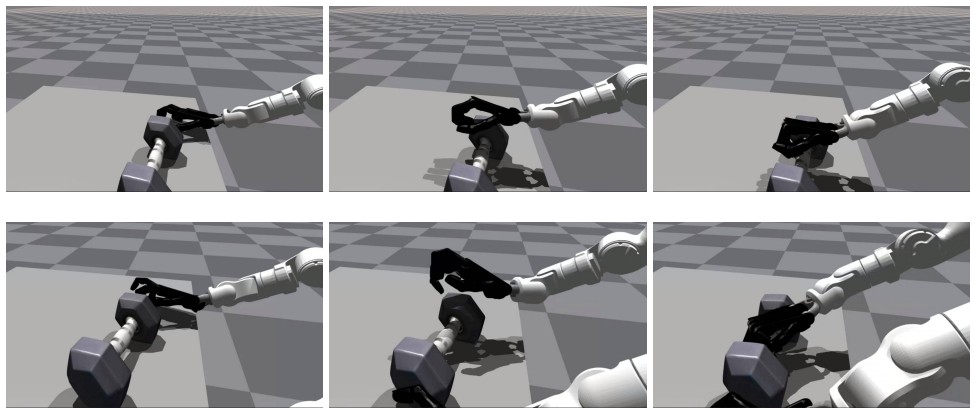

Figure 8: Failure (top row) and success (bottom row) cases for implausible dumb manipulation.

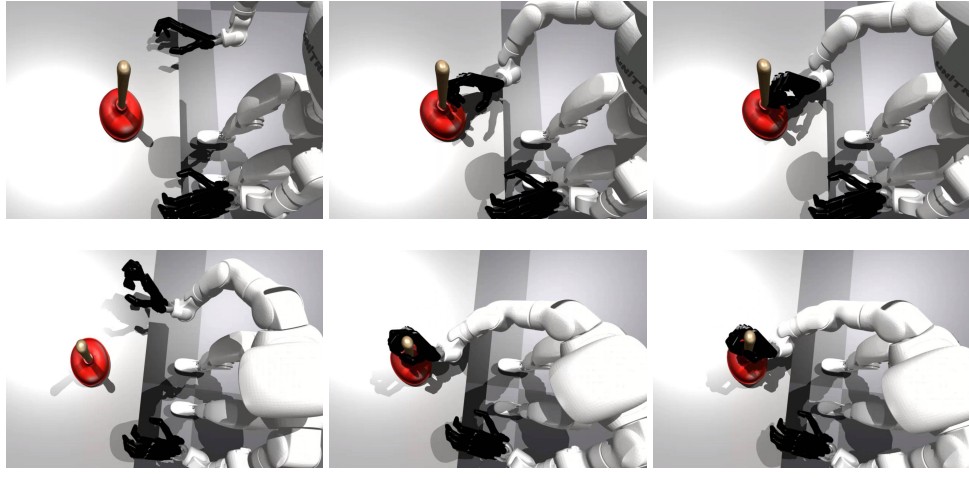

Figure 9: Failure (top row) and success (bottom row) cases for implausible plunger manipulation.

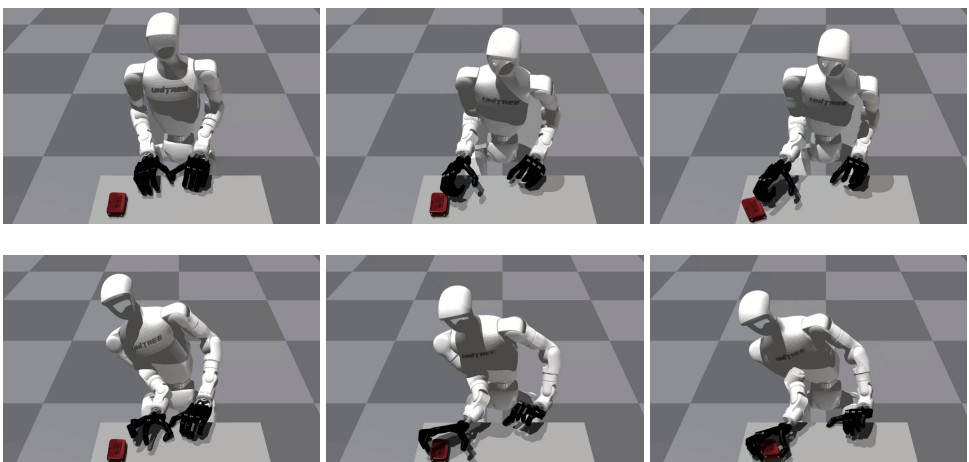

Figure 10: Failure (top row) and success (bottom row) cases for implausible eraser manipulation.

## C.2 FAILURE CASE ANALYSIS

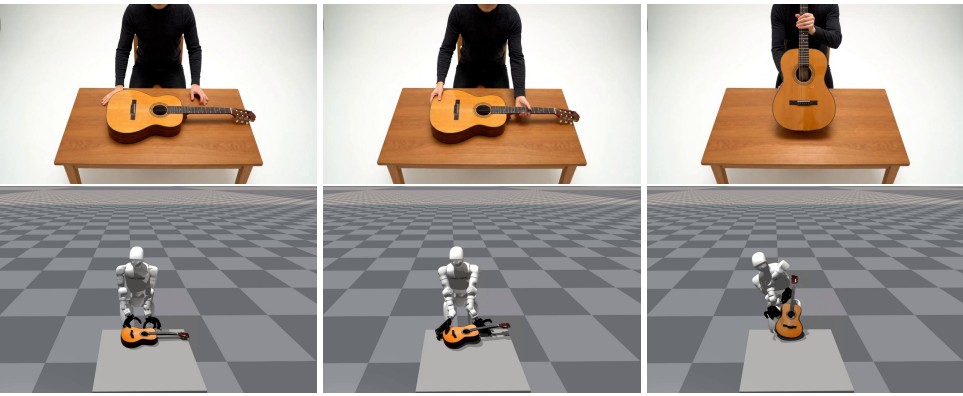

Figure 11: Failure case in the guitar-lifting task.

**Failure Case: Lifting a Guitar** One failure case occurs in the guitar-lifting task. The policy initially demonstrates promising behavior: the robot uses its right hand to stabilize the guitar body while the left hand supports the neck and lifts the instrument. Despite this impressive coordination, the task ultimately fails.

The failure stems from the mismatch between the robot's embodiment and the physical properties of the object. The guitar is both large and positioned close to the torso, which is manageable for a human but problematic for the Unitree G1. With a total height of only 132 cm, the robot's smaller scale forces its arms into awkward postures, preventing a stable grasp and leading to task failure.

This case underscores the embodiment gap between human demonstrations and robotic execution. While the policy can imitate human-like strategies, the limited size and morphology of the robot impose physical constraints that hinder the direct transfer of large-object manipulation skills.

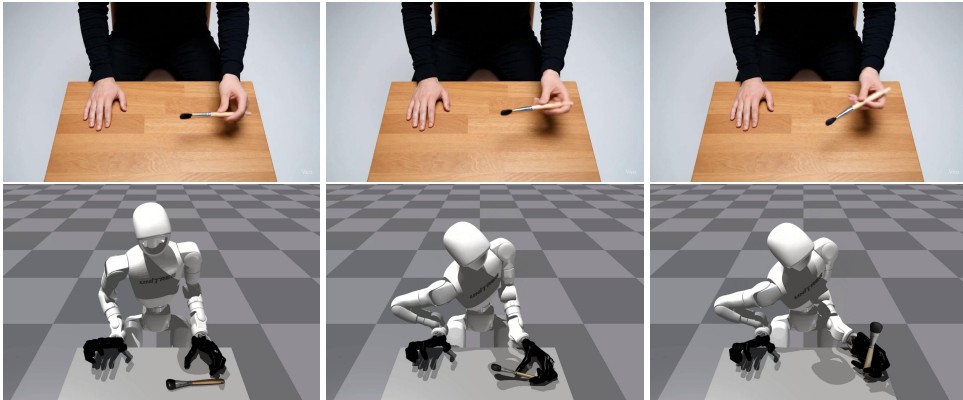

Figure 12: Failure case in the brush-picking task. Note that in the top row, second image, the brush visually penetrates the hand.

**Failure Case: Picking up a Brush**   A representative failure case appears in the brush-picking task. As illustrated in Fig. 12, the human demonstration depicts a left-hand grasp followed by a smooth in-hand rotation to reorient the brush into a writing posture. However, closer inspection of the generated video reveals that during the rotation phase the brush visually penetrates the demonstrator's hand, indicating that the video-generated trajectory is not strictly physically feasible.

Our RL policy partially reproduces the demonstration: it successfully grasps and lifts the brush, but fails to execute the rapid in-hand reorientation. This limitation highlights the dexterity gap between the human and the robot hand—where humans can seamlessly transition from grasping to fine-grained in-hand adjustments, the robot hand struggles to achieve the same level of precision.

Together, these observations reveal two key challenges. First, video-generated demonstrations, while visually plausible, may sometimes encode motions that are physically inconsistent or infeasible. Second, even when the demonstrations are feasible, tasks demanding quick and precise in-hand rotations expose fundamental limitations of current robot hand dexterity. This dual perspective underscores both the promise and the current limitations of video-based skill acquisition.

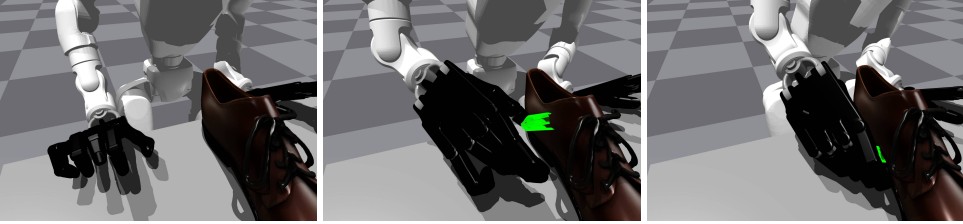

Figure 13: Failure case in the shoe-picking task.

**Failure Case: Picking up a Shoe**   Finally, in the shoe-pickup task, the policy attempted to pick up a shoe but was unable to establish a stable grasp. As shown, the robot hand approaches the shoe, yet the thumb makes contact on the outside surface rather than entering the shoe interior. This misalignment prevents the grasp from succeeding.

The limitation arises from the formulation of the contact reward. While the reward encourages the hand to move closer to the designated contact region, it does not explicitly distinguish whether the approach is made from the correct side of the object surface. As a result, the policy may converge to a local optimum where the thumb reduces the distance to the target region but establishes contact on the outside of the shoe, leading to an unsuccessful grasp.

This case highlights a limitation of the current reward design. In particular, attraction rewards should not only minimize geometric distance but also encourage contact with the correct object surface or

region. Incorporating such constraints could reduce spurious optima and improve the policy's ability to learn reliable grasp strategies for objects with concave or hollow geometries.

### C.3 POSE ESTIMATION ENHANCED BY POINT TRAJECTORIES

By combining FoundationPose with SpatialTracker, DexMan achieves more stable and accurate pose estimation. While vanilla FoundationPose often drifts under occlusion or fast motion, the additional trajectory guidance reduces jitter and preserves alignment across frames, as illustrated in Fig. 14.

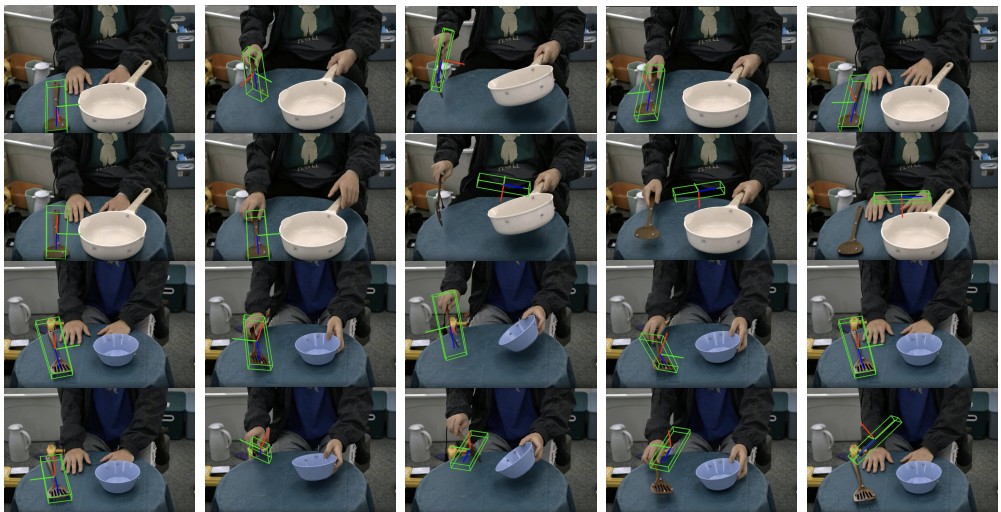

Figure 14: **Visualization of pose estimation results.** The first and third rows depict successful cases where DexMan, enhanced with SpatialTracker, outperforms vanilla FoundationPose. In contrast, the second and fourth rows highlight failure cases of vanilla FoundationPose.

### C.4 VIDEO-TO-ROBOT SKILL ACQUISITION

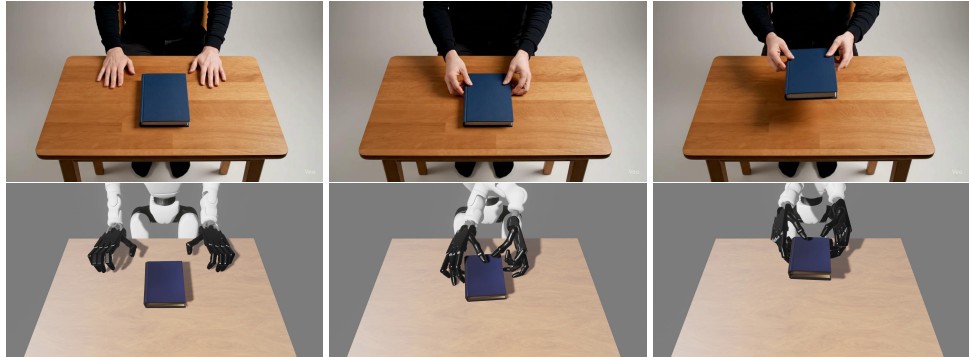

Figure 15: Success case in Veo3-generated video

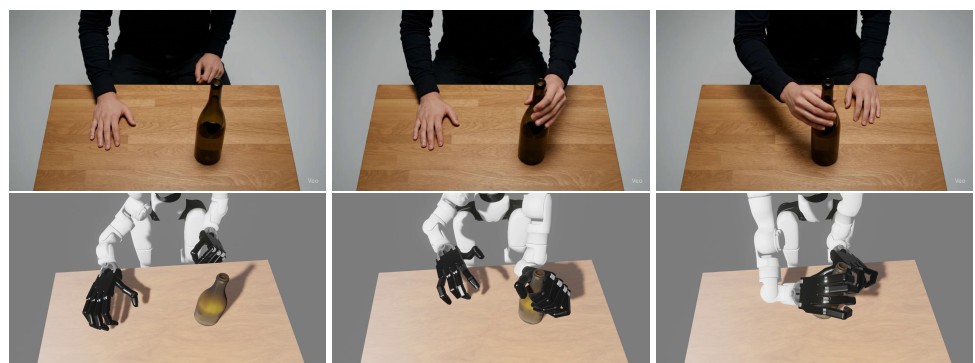

Figure 16: Success case in Veo3-generated video

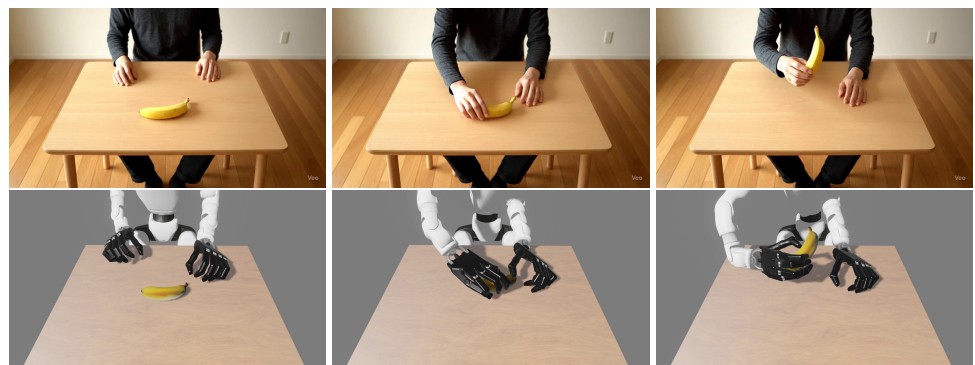

Figure 17: Success case in Veo3-generated video

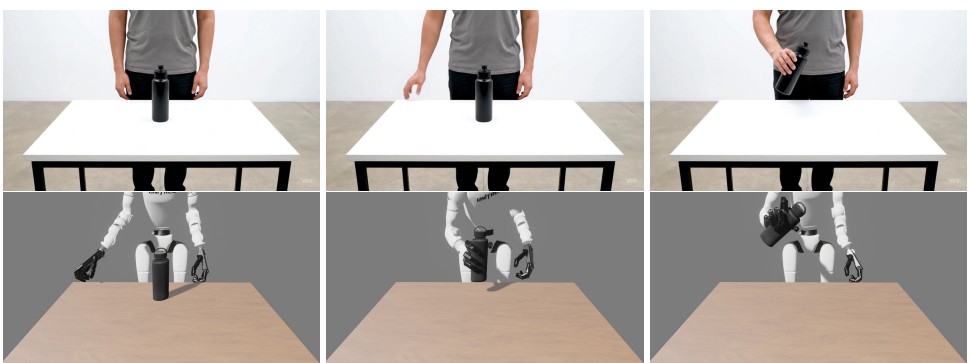

Figure 18: Success case in Veo3-generated video

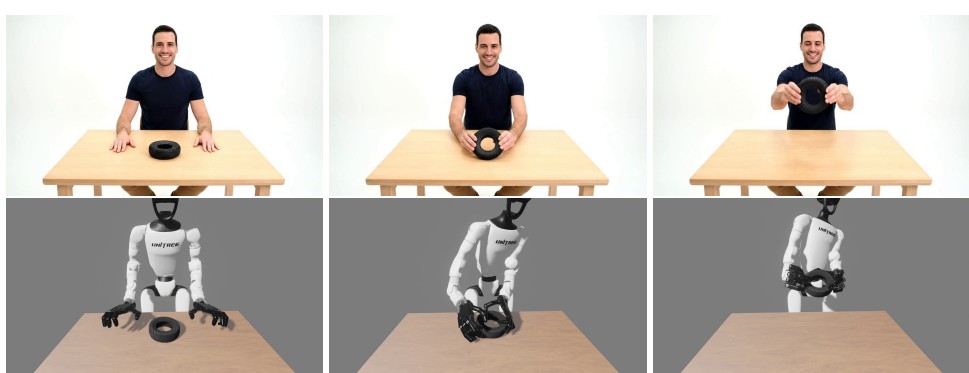

Figure 19: Success case in Veo3-generated video

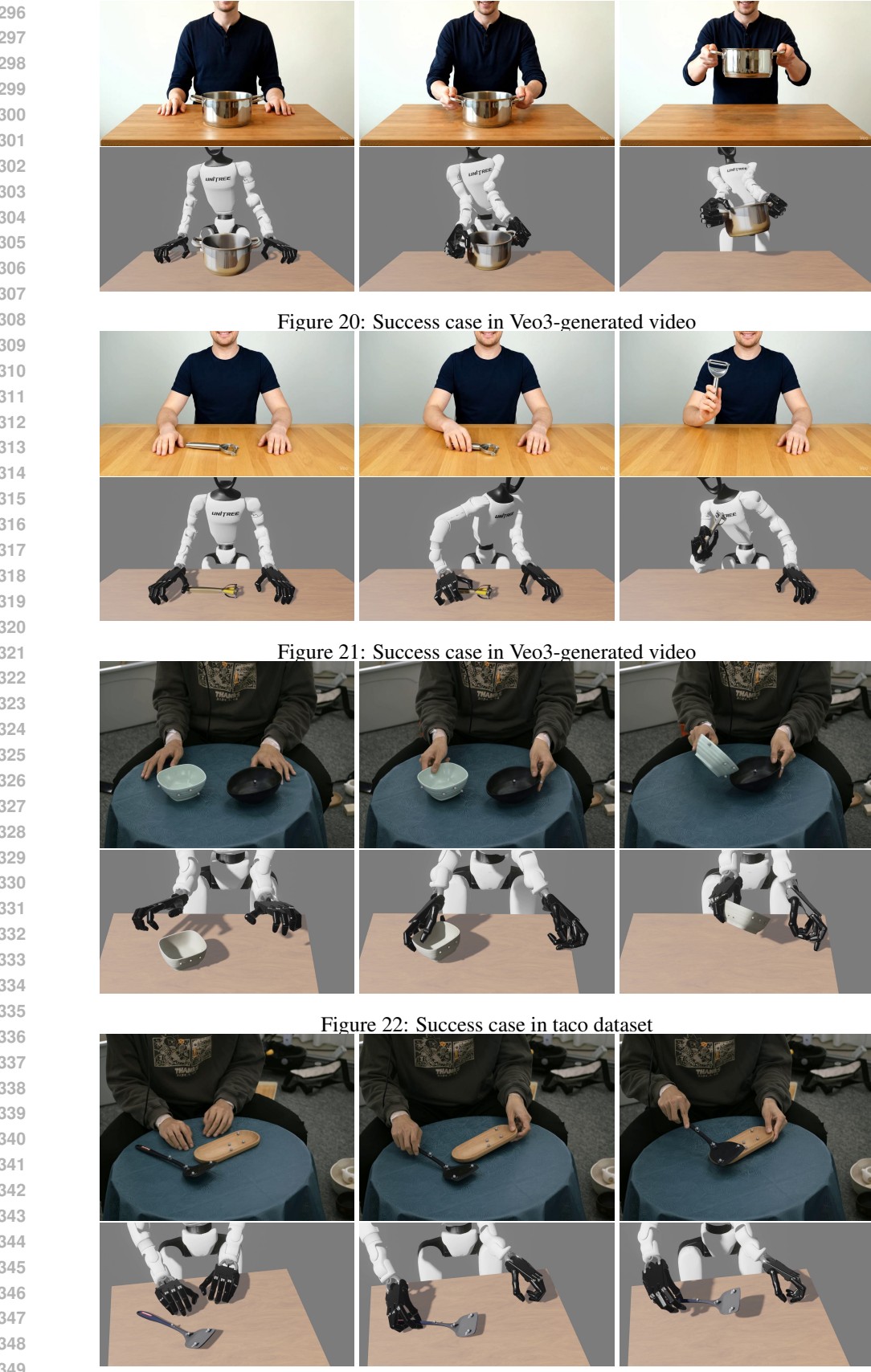

Figure 20: Success case in Veo3-generated video

Figure 21: Success case in Veo3-generated video

Figure 22: Success case in taco dataset

Figure 23: Success case in taco dataset

