# OpenReview forum: "DexMan: Learning Bimanual Dexterous Manipulation from Human and Generated Videos"
_ICLR.cc/2026/Conference — Submitted to ICLR 2026_

### Official Review · Reviewer_nTLv · 2025-10-24

**Soundness:** 2
**Presentation:** 3
**Contribution:** 2
**Rating:** 4
**Confidence:** 5

**Summary:**

DexMan is an automated “video-to-robot” pipeline that turns third-person monocular RGB videos of humans manipulating rigid objects into bimanual dexterous skills for a full humanoid robot—no camera calibration, depth sensors, scanned 3D assets, or ground-truth hand/object motion needed.

Contributions:

RGB-only → bimanual dexterity: First end-to-end pipeline that converts uncalibrated, third-person monocular videos (real or synthetic) into bimanual dexterous skills on a full humanoid in simulation.

Contact-prior attraction reward: Object-centric correspondence between hand keypoints and mesh vertices + normal alignment to encourage meaningful, stable grasps despite noisy video poses.

Pose estimation with motion cues: Fuses a 6D pose estimator (FoundationPose-style) with 3D point trajectory tracking to stabilize object pose across occlusion/fast motion.

Stable object placement: Sampling-and-simulate step that picks the closest stable configuration from imperfect reconstructions, improving RL training stability.



Conclusion:
The authors have done commendable work on this paper. However, the proposed method is relatively simple, and the reported success rate is below what I would expect—37% in simulation is not very compelling. In addition, the sim-to-real gap remains unaddressed due to the lack of real-world experiments. I strongly recommend including real-robot evaluations and reporting the corresponding results.

**Strengths:**

Robustness to noisy supervision: The contact-centric reward avoids trivial touches and contact-avoidance minima, markedly boosting success in ablations.

Bimanual + dexterous + humanoid: Tackles a much tougher setting than floating hands or single-arm grippers—evidence of strong system design.

Well-engineered perception stack: Depth normalization, scale estimation, pose refinement with tracked 3D trajectories—reduces temporal jitter and failure rate.

Trainability in sim: Stable-pose sampling and residual-on-retargeting control make PPO training feasible for high-DoF, contact-rich tasks.

**Weaknesses:**

Sim-only: No real-robot results; sim-to-real gap (contacts, friction, latency) remains unaddressed.

Scene scope: Single-human, rigid tabletop objects; no deformable/articulated objects or mobile manipulation.

Perception–contact decoupling: Hands/objects are estimated first, contacts inferred later; inconsistencies can propagate and misguide rewards.

Action parameterization: End-effector + finger residuals underuse full-arm posture optimization; potential self-collision/occlusion issues in clutter.

**Questions:**

Sim-to-real: What adaptations (domain randomization, contact model calibration, residual force control, tactile feedback) do you expect are most critical to transfer these policies to a physical humanoid?

Contact-prior extraction: How sensitive is performance to the distance thresholds (τᵢ) and vertex selection heuristics? Could a learned correspondence (e.g., contrastive point-cloud features) replace nearest-vertex?

Failure modes: For the ~60–70% unsuccessful video→skill cases, what dominates—pose drift, unstable reconstructions, grasp selection, or IK infeasibility? Any quantitative breakdown?

Articulated/deformable objects: What changes would be needed—object models, contact reward definition, or control parameterization—to handle doors, drawers, cloth?

---

> ### Author Response · Authors · 2025-11-27
> **We sincerely thank the reviewer for the thoughtful comments and look forward to addressing any remaining questions.**
>
> ---
>
> **Q1: Sim-to-real adaptation.**
>
> **A1**: Sim-to-real adaptation has different levels of complexity. From the simplest setup like ManipTrans [1], which directly replays successful demonstration trajectories in a simulator aligned to the real environment, the only requirement is that the replayed actions remain feasible on the real robot. To a more difficult setup where the real world and simulation is misaligned, we need to learn a generalist, robust policy that generalizes to a wide range of object properties, object geometries, hand-object dynamics, visual observations, and action control noise.  As a result, a huge suite of domain randomization should be carefully designed and used to train the robot policy.
>
> ---
>
> **Q2: Sensitivity of the contact prior & possibility of learning.**
>
> **A2**: Our contact reward aims to offer adaptive guidance, rather than hard constraints, on hand motion with respect to the current object pose.  As a result, it is not sensitive to the distance threshold and vertex selection heuristics.
>
> We agree that learning such structured correspondence from recently proposed hand-object contact dataset, such as Dex1b[2], would be a promising future work.
>
> ---
>
> **Q3: Detailed evaluation of the pipeline and success/failure analysis of the full pipeline.**
>
> **A3**: We added a table that reports end-to-end performance. In our experiments, we select videos where the objects are visible in the first frame for both TACO and Veo3-synthetic dataset, thereby, the object reconstruction success rate is 100%. The remaining failures typically arise from initial pose estimation, pose tracking, or reinforcement learning. For initial pose estimation, errors include wrong orientation, such as an upside-down initial pose. For pose tracking, the dominant issue is rotational drift. Reinforcement learning failures are discussed in the Appendix and are largely due to the limited action space of a fixed-base robot. We expect that these failures can be mitigated by introducing locomotion-manipulation.
>
> | **Dataset / Stage** | **Object Reconstruction** | **Initial Pose Estimation** | **Pose Tracking** | **RL Training** |
> |----------------------|---------------------------|------------------------------|--------------------|------------------|
> | **TACO**             | 100%                      | 78%                          | 64%                | 64%              |
> | **Veo3**             | 100%                      | 80%                          | 88%                | 60%              |
>
> ---
>
> **Q4: Extending to articulated / deformable objects.**
>
> **A4**: For manipulating articulated objects, a direct extension is to replace object-wise with part-wise segmentation and track the pose of each part, separately.  Since the articulation can be recovered from part-wise relative motions, the entire control pipeline remains unchanged. For manipulating deformable objects, there are two major bottlenecks, including (1) how to represent them geometrically, and (2) how to track particle-level motion robustly.  These two problems are still open research questions in both computer graphics and computer vision.  Otherwise, our three types of rewards–object following, hand imitation and contact reward, can be extended to both scenarios accordingly.
>
> ---
>
> **Weakness 2: Limited scene diversity.**
>
> **A5**: While this paper focuses on tabletop manipulation, our proposed method–DexMan, is not limited to this setting.  We can directly extend DexMan to mobile manipulation by adapting pure hand-imitation reward with that for whole-body motion imitation.
>
> ---
>
> **Weakness 3: Perception–contact decoupling.**
>
> **A6**: We agree this separation can be further improved. Both learning-based correspondence prediction and optimization-based refinement are natural future directions, and DexMan is flexible to incorporate these extensions.
>
> ---
>
> **Weakness 4: Action parameterization underutilizes full-arm optimization.**
>
> **A7**: DexMan is not meant to solve all aspects of whole-body and occlusion-aware control at once. Our goal is to provide a platform for exploring video-supervised dexterous manipulation. Future versions can incorporate richer action spaces, whole-body parameterizations, or occlusion-aware strategies without altering the overall pipeline.
>
> ---
>
> ### References
>
> [1] Li, et al. ManipTrans: Efficient Dexterous Bimanual Manipulation Transfer via Residual Learning, 2025.
>
> [2] Ye, et al. Dex1B: Learning with 1B Demonstrations for Dexterous Manipulation, 2025.

---

> ### Comment · Reviewer_nTLv · 2025-11-27
> **Thanks for your feedback**
>
> I think most of my concern has been addressed and i agree with your statement.
>
> But for the sim2real deployment, and this statement, i cannot agree with that: "Maniptrans: which directly replays successful demonstration trajectories in a simulator aligned to the real environment".
>
> In this paper,you are using Isaacgym, which has a very large sim2real gap for dexterous manipulation due to its simulation backbone setting. That is why we normally require the sim2real experiment for most of the top-tier robotic learning paper in conference like iclr, neurips and rss. For example, Isaacgym do not hold for the conservation of angular momentum, which may lead to significant sim2real gap when manipulate object with wired geometry or non-uniform mass distribution.
>
> The new nvidia solver: Newton is designed to fix this issue but it is still under developed.
> We hope that one day, the sim2real gap can be eliminated and we can use the simulation benchmark for real world task.
> But i am sorry that in the 2025, this is still not possible.
> I believe that if you can show some real world experiment result, most of the reviewers are willing to improve our score. But given the natural of ICLR as a learning conference. I just cannot give you a positive score with only simulation result.

---

> > ### Author Response · Authors · 2025-11-27
> >
> > We appreciate the reviewer’s clarification regarding sim-to-real challenges. However, we respectfully note that our original statement about ManipTrans was accurate in the context of how the method itself describes its real-robot deployment. Specifically, the authors of ManipTrans explicitly confirmed the following in their public issue discussion:
> >
> > > *“To make sure we’re aligned: as described in the paper, our real-world experiments replay precomputed trajectories. In ManipTrans, the real-robot phase does not use visual perception or tactile sensing, so these modalities do not affect the reported success rate.”*
> > > — ManipTrans authors, GitHub Issue #49 [1]
> >
> > This official clarification indicates that their real-robot results rely on direct replay of precomputed trajectories, not on online perception or contact-rich feedback control. In addition, ManipTrans is also implemented in IsaacGym, meaning that the same simulation-physics limitations mentioned by the reviewer apply equally to that system as well.
> >
> > We agree that real-robot experiments are important evidence for transfer. Due to the lack of access to appropriate hardware at this stage, we are not able to carry out the experiments, and we fully understand that the lack of real-robot validation may limit how much the reviewer is willing to adjust the score. We appreciate the reviewer’s feedback, and will pursue sim-to-real experiments in future extensions of this work.
> >
> > ---
> > ### References
> > [1] https://github.com/ManipTrans/ManipTrans/issues/49

---

### Official Review · Reviewer_4mRD · 2025-10-29

**Soundness:** 2
**Presentation:** 3
**Contribution:** 2
**Rating:** 4
**Confidence:** 4

**Summary:**

The paper proposes a pipeline that converts a third-person human manipulation video to a robotic manipulation policy in simulation. First, many off-the-shelf vision models are used to estimate object meshes, object poses, and hand poses from the video. Then, the estimated hand-object interaction data serves as the task goal in simulation and a policy is trained via RL to control a humanoid upper-body to complete the same task. Experiments show that the proposed RL pipeline outperforms ManipTrans, and the whole pipeline can achieve successful policy learning from AI generated videos.

**Strengths:**

- While many prior works study learning robotic policies from mocap data, learning from pure videos is a novel challenge. This paper is a good initial exploration in this direction.
- The paper writing is easy to follow. Implementation of the pipeline is clear.

**Weaknesses:**

- My major concern is that, the problem studied in this paper (video-to-policy) is actually divided into two distinct research problems (1. video-to-mocap-data and 2. mocap-data-to-policy). While the paper studies both, for the first problem, the paper has not analyze the accuracy of estimated poses and object meshes and compare with any prior methods; for the second problem, the paper only compares with ManipTrans but ignores many related approaches [1,2,3,4] that learn RL policies from mocap manipulation data.
- For the first video-to-mocap task stage, the method introduces a lot of off-the-shelf models (e.g. depth estimation, semantic segmentation, hand pose estimation, and 3D reconstruction). Such a combined approach will introduce compounding errors to the system. The paper has not analyze errors introduced by each module and present accuracy and failure cases of this task stage.
- According to figures and the website, most tasks are simple pick-and-place tasks, and all tasks from AI generated videos are grasping tasks. It is questionable whether the system works for complex contact-rich manipulation tasks and the video generation model can correctly generate these videos.
- The experimental settings have not consider the sim-to-real potential. The Shadow Hands are simplified, removing the large cylinder arms. The learned behavior has many robot-table collisions according to the videos.

[1] Chen et al., Object-Centric Dexterous Manipulation from Human Motion Data, 2024
[2] Zhou et al., Learning Diverse Bimanual Dexterous Manipulation Skills from Human Demonstrations, 2024.
[3] Gao et al., CooHOI: Learning Cooperative Human-Object Interaction with Manipulated Object Dynamics, 2024
[4] Xu et al., InterMimic: Towards Universal Whole-Body Control for Physics-Based Human-Object Interactions, 2025

**Questions:**

- How many tasks from the TACO dataset, human videos, and AI generated videos are selected for experiments? What is the selection criterion? Which module in the whole pipeline causes failure cases?
- Are object initial states and robot initial poses randomized for RL training?
- Please refer to the Weaknesses above.

---

> ### Author Response · Authors · 2025-11-23
> **We sincerely thank the reviewer for the thoughtful comments and look forward to addressing any remaining questions.**
>
> ---
>
> **Q1: Dataset selection and failure analysis of the full pipeline.**
>
> **A1**: We use TACO pre-release dataset and filter out clips where the human is not visible or objects are completely occluded in the first video frame.  We sample 50 videos based on this criteria. For AI-generated videos, we discard clips with invisible humans or moving cameras and collect 50 qualified videos.
>
> ---
>
> **Q2: Are object/robot initial states randomized during RL?**
>
> **A2**: We did not randomize the object or robot initial states during RL in our experiments because the goal of this paper is to reconstruct specific robot skills directly from video, rather than to train a generalist policy. Developing a generalist, robust policy is part of our future work, where we will apply randomization to both object and robot initial states. Our contact-conditioned reward is designed to preserve meaningful hand–object relations under such perturbations.

---

> > ### Author Response · Authors · 2025-11-23
> >
> > ---
> >
> > **Weakness1. No quantitative evaluation of video-to-mocap accuracy or comparisons to prior perception methods.**
> >
> > **A3**: Our contribution is the end-to-end bridge from third-person videos to policy with contact-aware objectives that make RL tolerant to noisy hand/object estimates
> > For object reconstruction, we use Trellis off-the-shelf; consequently, we do not include horizontal comparisons across mesh generation methods. For pose/hand tracking, we enhance a state-of-the-art pose estimation model–FoundationPose, with 3D point tracking.  We conduct an ablation study to analyze the effectiveness of our enhancement.
> >
> > For the papers suggested by the reviewer, they adopt very different settings from ours, including unimanual jaw-based manipulation vs. bimanual dexterous manipulation, end-effector pose control vs. joint angle control, loco-manipulation vs. tabletop manipulation, image-based policy vs. state-based policy.  As a result, a thorough and fair comparison is not plausible given the limited time.  We will clarify differences in learning objectives and experimental settings in the paper.  Meanwhile, we will compare the effectiveness of our proposed reward objectives against theirs to highlight our contribution.
> >
> > **Weakness 2. No end-to-end evaluation or discussion of pipeline fragility.**
> > **A4**: We added a table that reports end-to-end performance. In our experiments, we select videos where the objects are visible in the first frame for both TACO and Veo3-synthetic dataset, thereby, the object reconstruction success rate is 100%. The remaining failures typically arise from initial pose estimation, pose tracking, or reinforcement learning. For initial pose estimation, errors include wrong orientation, such as an upside-down initial pose. For pose tracking, the dominant issue is rotational drift. Reinforcement learning failures are discussed in the Appendix and are largely due to the limited action space of a fixed-base robot. We expect that these failures can be mitigated by introducing locomotion-manipulation.
> >
> > | **Dataset / Stage** | **Object Reconstruction** | **Initial Pose Estimation** | **Pose Tracking** | **RL Training** |
> > |----------------------|---------------------------|------------------------------|--------------------|------------------|
> > | **TACO**             | 100%                      | 78%                          | 64%                | 64%              |
> > | **Veo3**             | 100%                      | 80%                          | 88%                | 60%              |
> >
> >
> > **Weakness 3.  Limited task complexity**
> >
> > **A5**: In the TACO dataset, we include other types of manipulation tasks, such as using a roller brush or stirring with a ladle. Beyond simple pick-and-place, tasks involving large-area in-hand contacts (e.g., screw-cap motions) remain challenging: current video-generation models often occlude the object, which degrades object tracking. This limitation affects our pipelines when reliable object dynamics cannot be guaranteed. That said, given the rapid pace of progress in video generative models, we expect upcoming releases to significantly reduce these issues.
> >
> > **Weakness 4. Limited sim-to-real relevance due to simplified hardware and collisions.**
> >
> > **A6**:  We'd like to highlight that our method is not restricted to any type of dexterous robot hand.  In this paper, following prior work, such as ManipTrans[1] / UniDexGrasp[2] / DexGraspNet[4], we remove the forearm cylinder from the Shadow Hand, and we focus on learning high-DoF action control in simulation.  We aim to answer if contact-aware objectives enable fast exploration in high-DoF reinforcement learning.
> >
> > Sim2real transfer in bimanual dexterous manipulation is still an open research question.  A recent paper VIRAL[4], shows that, in order to reduce sim2real gap, scaling up RL training in simulation is necessary. While it requires a tremendous amount of effort to tackle sim2real transfer, and we are committed to address sim2real transfer in the next step.
> >
> > ---
> > ### References
> >
> > [1] Li, et al. ManipTrans: Efficient Dexterous Bimanual Manipulation Transfer via Residual Learning, 2025.
> >
> > [2] Xu, et al. UniDexGrasp: Universal Robotic Dexterous Grasping via Learning Diverse Proposal Generation and Goal-Conditioned Policy, 2023.
> >
> > [3] Wang, et al. DexGraspNet: A Large-Scale Robotic Dexterous Grasp Dataset for General Objects Based on Simulation, 2022.
> >
> > [4] He, et al. VIRAL: Visual Sim-to-Real at Scale for Humanoid Loco-Manipulation, 2025.

---

### Official Review · Reviewer_fmaV · 2025-10-30

**Soundness:** 2
**Presentation:** 3
**Contribution:** 2
**Rating:** 4
**Confidence:** 3

**Summary:**

This paper presents DexMan, an automated framework that learns bimanual dexterous manipulation skills for humanoid robots directly from third-person human demonstration videos. Unlike previous approaches requiring calibrated cameras, depth data, or motion capture, DexMan operates entirely on unannotated videos and introduces contact-based reward functions to improve policy learning from noisy hand–object pose estimates. It achieves state-of-the-art results in object pose estimation on the TACO dataset and outperforms prior reinforcement learning methods on OakInk-v2 by 19% in success rate. Furthermore, DexMan can learn from both real and synthetic videos, eliminating the need for manual data collection and enabling the creation of large-scale, diverse datasets to train generalist dexterous manipulation policies.

**Strengths:**

1. The paper proposes a pipeline that can learn bimanual dexterous manipulation skills in simulation from a third-person monocular human video.
2. The paper is well-written, presenting a complex technical system with conceptual clarity and a logical narrative that is easy to follow.

**Weaknesses:**

1. The experimental results are limited to simulation and lack sim-to-real experiments.
2. The novelty of the proposed method is limited. The object and hand pose estimation parts leverage some off-the-shelf methods for hand pose estimation and object reconstruction. The policy learning part also uses some common architecture and reward designs similar to [1], [2], [3], [4], [5].

[1] Li K, Li P, Liu T, et al. Maniptrans: Efficient dexterous bimanual manipulation transfer via residual learning[C]//Proceedings of the Computer Vision and Pattern Recognition Conference. 2025: 6991-7003.

[2] Mandi Z, Hou Y, Fox D, et al. Dexmachina: Functional retargeting for bimanual dexterous manipulation[J]. arXiv preprint arXiv:2505.24853, 2025.

[3] Chen Y, Wang C, Yang Y, et al. Object-centric dexterous manipulation from human motion data[J]. arXiv preprint arXiv:2411.04005, 2024.

[4] Yuan Z, Wei T, Gu L, et al. Hermes: Human-to-robot embodied learning from multi-source motion data for mobile dexterous manipulation[J]. arXiv preprint arXiv:2508.20085, 2025.

[5] Lin, Toru, et al. "Sim-to-real reinforcement learning for vision-based dexterous manipulation on humanoids." arXiv preprint arXiv:2502.20396 (2025).

**Questions:**

1. In the ablation on reward components, you mention that contact rewards can be more crucial than task rewards. I'm curious whether “task reward” here refers to only object-following reward, only imitation reward, or both.
2. I am wondering whether high-quality object assets can be acquired directly from Trellis, since the extracted mesh should have similar shapes to real objects, as well as have plausible collision for contact-rich dexterous tasks.
3. How can you align the scales of object and hand meshes with real-world settings? It seems that the pipeline does not have access to indicators of real scale, such as camera extrinsic or real point clouds observation.

---

> ### Author Response · Authors · 2025-11-23
> **We sincerely thank the reviewer for the thoughtful comments and look forward to addressing any remaining questions.**
>
> ---
>
> **Q1: What does “task reward” mean in the ablation?**
>
> **A1**: “Task reward” refers only to the object-following term.
>
> ---
>
> **Q2: Can Trellis assets be used directly for contact-rich manipulation?**
>
> **A2**: This is a good question. While Trellis produces high-fidelity meshes with detailed geometry over most visible regions, perfect reconstruction is impossible and can indeed affect contact dynamics. This is precisely why we use RL: even if the reconstructed object differs from ground truth, the policy still learns stable grasps and robust manipulation behaviors.
>
> ---
>
> **Q3: How are real-world scales recovered without calibration or metric depth?**
>
> **A3**: As illustrated in Appendix B.2, we infer scale from the reconstructed human hand model. We use HaMeR, which reconstructs hand mesh in metric scale (close to actual human-hand size). We use this prior to scale VGGT’s metrically incorrect depth so that the hand and depth are aligned in scale, yielding a globally correct metric depth. We then use Any6D over the predicted metric depth to recover the correct object scale, making the final mesh scale consistent with the scene.
>
> ---
>
> **Weakness 2: Novelty of the contact reward.**
>
> **A4**: While we adopt a common combination of object-following, hand-imitation and contact reward, we want to emphasize that our novel lies in the design of the contact reward. Previous approaches to contact reward either ignore structured correspondence of hand-object contact or require task-specific tuning.  We'll distinguish between our contact reward and others as follows.
>
> *ManipTrans*[1]: it rewards the nearest distance from the hand to any point on the object. This does not guide the hand to the right surfaces for the task; with noisy reconstructions it often converges to “touch somewhere” rather than positions that induce stable grasp.
>
> *Sim-to-real RL for humanoids*[2]: it uses manually defined contact markers. This method is limited in three aspects: (i) needs per-task annotation, and (ii) produces fixed contact points over time, so the policy learns one static grasp instead of learning when to make, release, or shift contacts; (iii) It does not prevent back-of-hand contact which results in infeasible grasp.
>
> *HERMES*[3]: It only attracts the fingers toward the object's geometric center.  However, without considering where to contact the object with which finger, getting close to the center only doesn't ensure feasible grasp.
>
> *DexMachina*[4]: It calculates the absolute 3D positions of contact points.  In other words, it enforces the model to follow a fixed trajectory, failing to adapt to varying object poses during training.
>
> *Object-centric dexterous manipulation from human motion data*[5]: it focuses on object following and imitation reward, while excluding contact reward.
>
> In stark contrast, our contact rewards jointly considers (i) proximity to task-relevant regions, (ii) surface normal alignment to prevent back-of-hand contacts, and (iii) a time-varying, video-derived contact affordance. Our proposed contact reward can be automatically derived from the video, without any manual tuning for each task, and this structured correspondence adapts contact positions based on the current object pose.
>
> ---
>
> ### References
>
> [1] Li, et al. ManipTrans: Efficient Dexterous Bimanual Manipulation Transfer via Residual Learning, 2025
>
> [2] Lin, et al. Sim-to-Real Reinforcement Learning for Vision-Based Dexterous Manipulation on Humanoids, 2025
>
> [3] Yuan, et al. HERMES: Human-to-Robot Embodied Learning from Multi-Source Motion Data for Mobile Dexterous Manipulation, 2025
>
> [4] Zhao, et al. DexMachina: Functional Retargeting for Bimanual Dexterous Manipulation, 2025.
>
> [5] Chen, et al. Object-Centric Dexterous Manipulation from Human Motion Data, 2024.

---

> > ### Comment · Reviewer_fmaV · 2025-11-27
> >
> > Thank you to the authors' replies! I still have several concerns:
> > 1. The experimental results are limited to simulation and lack sim-to-real experiments, which is mentioned in the previous review phase.
> > 2. I'm quite confused by the answer to Q2. It seems that you train a separate policy for each task without category-level object randomization. If your extracted object mesh differs from the ground truth in terms of collision or even shape, the learned policy will overfit to this "imperfect" object. How can it learn "stable grasps and robust manipulation behaviors"?
> > 3. Can you provide some quantitative evaluation results on the method you mentioned in A3? For example, what is the error between the estimated scales of objects and the ground truths?

---

> ### Author Response · Authors · 2025-11-27
>
> We now understand the reviewer’s point clearly. For an individual task aiming at real-world transfer, the reconstructed object in simulation must match the real object with high geometric fidelity in both shape and scale; without such alignment, transferring a single-task policy is naturally very challenging. But the downside of this per-task focus is that the trained policy would primarily handle only that exact object; accordingly, this work does not center on per-task sim-to-real transfer.
>
>
> Instead, our goal is to learn a large number of manipulation skills directly from videos. Even though each individual task may not perfectly reproduce the real object geometry, training on many diverse tasks exposes the policies to a wide variety of object shapes, scales, and reconstruction imperfections. Our speculation is that, with sufficiently large data, the model may begin to capture common manipulation patterns across different objects and potentially develop a degree of generalization to new shapes. However, we have not yet trained at the scale required to fully validate this hypothesis, nor have we built a generalist policy or conducted sim-to-real transfer experiments. These remain important directions for future work.
>
>
> Regarding the reviewer’s concern about object scale accuracy, we agree that this is an important factor. As suggested, we will provide additional quantitative evaluation on TACO objects to measure scale estimation error relative to ground-truth object dimensions.

---

### Official Review · Reviewer_m6pD · 2025-11-03

**Soundness:** 3
**Presentation:** 3
**Contribution:** 3
**Rating:** 6
**Confidence:** 3

**Summary:**

This paper presents an approach that can help convert human videos into trajectories for robotic learning of bimanual dexterous manipulation skills. The approach reconstructs the 3D hand and object motion from video and trains an RL policy that reproduces this sequence. This functionality is demonstrated using both real and generated videos.

**Strengths:**

- I appreciate the effort of the paper to offer an end to end pipeline with all the pieces to learn a robotic skill from videos of human demonstrations.
- When components of the approach are evaluated independently, they achieve better performance than recent baselines (Tables 1 and 2).
- There is clear care in getting a sensible solution for 3D hand and object motion recovery and a robust approach for retargeting and RL policy training.
- The main paper and the supplementary do a good job presenting enough details for the approach and the implementation details.

**Weaknesses:**

- Table 1 performs an evaluation assuming that the 3D model of the object is provided. The rest of the proposed pipeline (e.g., 3D object reconstruction) is not considered in the evaluation.
- The evaluation only considers independent components of the approach. It would be helpful to factor in all steps. The pipeline is quite elaborate, so it could lead to a fragile method. This is not discussed properly.

**Questions:**

- I would be interested in seeing a more detailed evaluation of the pipeline and success/failure analysis of all the steps for general videos of human demonstration.
- How does the method deal with occlusions for the object when applying the object reconstruction network (Trellis)?
- Were other networks/approaches considered for the hand/object reconstruction part?

---

> ### Author Response · Authors · 2025-11-23
> **We sincerely thank the reviewer for the thoughtful comments and look forward to addressing any remaining questions.**
>
> ---
>
> **Q1: Detailed evaluation of the pipeline and success/failure analysis of the full pipeline.**
>
> **A1**: We added a table that reports end-to-end performance. In our experiments, we select videos where the objects are visible in the first frame for both TACO and Veo3-synthetic dataset, thereby, the object reconstruction success rate is 100%. The remaining failures typically arise from initial pose estimation, pose tracking, or reinforcement learning. For initial pose estimation, errors include wrong orientation, such as an upside-down initial pose. For pose tracking, the dominant issue is rotational drift. Reinforcement learning failures are discussed in the Appendix and are largely due to the limited action space of a fixed-base robot. We expect that these failures can be mitigated by introducing locomotion-manipulation.
>
> | **Dataset / Stage** | **Object Reconstruction** | **Initial Pose Estimation** | **Pose Tracking** | **RL Training** |
> |----------------------|---------------------------|------------------------------|--------------------|------------------|
> | **TACO**             | 100%                      | 78%                          | 64%                | 64%              |
> | **Veo3**             | 100%                      | 80%                          | 88%                | 60%              |
>
> ---
>
> **Q2: Performance of the object reconstruction network under occlusions.**
>
> **A2**: Trellis degrades under heavy occlusion. It may reconstruct parts of the hand, or it may generate imperfect objects with holes even after we mask the hand. Amodal3R[1] modifies Trellis to better handle occlusion and can improve object completion. However, even when the object is reconstructed, the initial pose estimator often fails under heavy occlusion. Recently, an image-to-3D reconstruction foundation model – SAM3D[2] was proposed.  We believe such a framework can further improve our video-to-robot skill acquisition pipeline.
>
> ---
>
> **Q3: Other networks/approaches considered for hand/object reconstruction.**
>
> **A3**: Most existing hand–object 3D reconstruction methods [3][4][5][6] only consider reconstruction from a single video frame where the hand and object are in contact.  These methods heavily depend on such assumptions to optimize hand and object poses. As a result, they are not suitable in our pipeline: we need to reconstruct the hand and object mesh consistently across the whole video, meanwhile, we do not assume constant hand-object contact in our use case.
>
> Instead, we consider long-horizon videos where humans reach, grasp, then move the object.  While video reconstruction approaches [7] exist, they are restricted: (1) they can only handle short video clips (typically less than five seconds), (2) they still the hand is in contact with the object, and (3) their output object models are often more sub-optimal than those produced by Trellis.
>
> While these methods underperform Trellis, we can still borrow their ideas to improve our pipeline.  For example, we can perform optimization or diffusion-guided refinement to improve hand–object reconstruction and reduce noise from hand/object pose estimation.
>
> ---
>
> **Weakness1: The rest of the pipeline component untested.**
>
> **A4**: Table 1 performs an evaluation assuming that the 3D model of the object is provided. The rest of the proposed pipeline is not considered in the evaluation. Please refer to A1 for the object reconstruction part.
>
> ---
>
> **Weakness 2: No end-to-end evaluation or discussion of pipeline fragility.**
>
> **A5**: Please refer to A1 for the end-to-end analysis.
>
> ---
>
> ### Referencences
>
> [1] Wu, et al. Amodal3R: Amodal 3D Reconstruction from Occluded 2D Images, 2025.
>
> [2] Chen, et al. SAM 3D: 3Dfy Anything in Images, 2025.
>
> [3] Tzionas, et al. 3D object reconstruction from hand-object interactions, 2015.
>
> [4] Tekin, et al. Unified egocentric recognition of 3D hand-object poses and interactions, 2019.
>
> [5] Zhang, et al. Perceiving 3D human-object spatial arrangements from a single image in the wild, 2020.
>
> [6] Zhang, et al. MOHO: Learning Single-view Hand-held Object Reconstruction with Multi-view Occlusion-Aware Supervision, 2024.
>
> [7] Liu, et al. EasyHOI: Unleashing the Power of Large Models for Reconstructing Hand-Object Interactions in the Wild, 2025.

---

### Author Response · Authors · 2025-12-04
**Overall Response and Appreciation**

We sincerely thank all reviewers for their thoughtful and constructive feedback. We appreciate the time and expertise invested in evaluating our work, and your comments have helped us substantially clarify the scope, limitations, and contributions of DexMan.

## Strengths Identified Across Reviews
Across the reviews, we are grateful that the following strengths of our work were recognized:
- Novel end-to-end pipeline that converts third-person monocular videos --real or synthetic -- into bimanual dexterous manipulation skills for humanoid robots. (m6pD, fmaV, 4mRD, nTLv)
- Robust perception stack, including scale-corrected depth normalization, stabilized pose estimation, and object reconstruction, with ablations demonstrating effectiveness. (m6pD, nTLv)
- Contact-prior reward that provides structured, adaptive hand–object correspondence and significantly improves training stability under noisy supervision. (m6pD, nTLv)
- Challenging setting, combining bimanual control, dexterous fingers, and humanoid kinematics rather than floating hands or simplified grippers. (nTLv)

## Main Concerns and Responses
We summarize key concerns from reviewers and outline how we addressed each in the rebuttal:

### **Lack of sim-to-real experiments**
We clarified the scope of this work: our focus is scaling video-to-skill acquisition, not per-task sim-to-real transfer. Nevertheless, we discussed realistic pathways to sim-to-real (domain randomization, contact calibration, generalist policy training). We acknowledge this as important future work.

### **Pipeline fragility & missing end-to-end evaluation**
We added a full stage-by-stage success table for both TACO and Veo3-synthetic, identifying failure modes in object reconstruction, initial pose estimation, pose tracking, and RL.

### **Novelty of the contact reward**
We expanded comparison against ManipTrans, DexMachina, HERMES, and object-centric methods, showing that our reward uniquely provides time-varying, surface-normal-aware, correspondence-based contact guidance derived automatically from video.

### **Limited task diversity / complexity**
We highlighted non-pick-and-place tasks in TACO and provided a detailed discussion of current limitations of video-generated content, and how the pipeline can naturally extend to articulated objects.

### **Comparison to additional mocap-based works**

We clarified differences in problem formulation, embodiment, and action space that make direct comparison difficult. We added a commitment to compare reward components against these methods in the revision.

---

### Meta-Review · Area_Chair_ZUYM · 2025-12-25

**Summary:**

The paper presents DexMan, a system for learning robotic manipulation skills from human videos. The approach first reconstructs 3D hand–object shape and motion from human videos, and then trains a RL policy using new contact-based rewards.

The initial scores are 6, 4, 4, and 4. Reviewers generally appreciated the engineering effort in building this complex pipeline and the clarity of the paper’s presentation.

However, several significant concerns were consistently raised:

1. Insufficient analysis of the full pipeline (m6pD, 4mRD).

2. Limited methodological novelty, as most components rely on off-the-shelf models, and the policy learning stage adopts reward designs and architectures similar to prior work. Moreover, there are insufficient comparisons with prior works for the two stages (fmaV, 4mRD).

3. Simple environments and tasks (4mRD, nTLv).

4. Lack of sim-to-real transfer (fmaV, 4mRD, nTLv).

In the rebuttal, the authors addressed the first concern, but the responses to the other concerns were limited.

For a leading learning conference, stronger methodological contributions, more thorough ablations, and more comprehensive experimental comparisons are expected. Therefore, the AC recommends rejecting the paper.

**Reviewer Concerns:**

See above.

**Reviewer Scores:**

The reviewers' score can remain the same.

---

### Decision · Program_Chairs · 2026-01-26

Reject